# The horizontally-acquired response regulator SsrB drives a *Salmonella* lifestyle switch by relieving biofilm silencing

Stuti K Desai[1], Ricksen S Winardhi[1,2], Saravanan Periasamy[3], Michal M Dykas[4,5], Yan Jie[1,2], Linda J Kenney[1,6,7]*

[1]Mechanobiology Institute, National University of Singapore, Singapore, Singapore; [2]Department of Physics, National University of Singapore, Singapore, Singapore; [3]Singapore Centre on Environmental Life Sciences Engineering, Nanyang Technological University, Singapore, Singapore; [4]Nanoscience and Nanotechnology Institute, National University of Singapore, Singapore, Singapore; [5]Graduate School for Integrative Sciences and Engineering, National University of Singapore, Singapore, Singapore; [6]Jesse Brown Veterans Affairs Medical Center, University of Illinois-Chicago, Chicago, United States; [7]Department of Microbiology and Immunology, University of Illinois-Chicago, Chicago, United States

**Abstract** A common strategy by which bacterial pathogens reside in humans is by shifting from a virulent lifestyle, (systemic infection), to a dormant carrier state. Two major serovars of *Salmonella enterica*, Typhi and Typhimurium, have evolved a two-component regulatory system to exist inside *Salmonella*-containing vacuoles in the macrophage, as well as to persist as asymptomatic biofilms in the gallbladder. Here we present evidence that SsrB, a transcriptional regulator encoded on the SPI-2 pathogenicity-island, determines the switch between these two lifestyles by controlling ancestral and horizontally-acquired genes. In the acidic macrophage vacuole, the kinase SsrA phosphorylates SsrB, and SsrB~P relieves silencing of virulence genes and activates their transcription. In the absence of SsrA, unphosphorylated SsrB directs transcription of factors required for biofilm formation specifically by activating *csgD* (*agfD*), the master biofilm regulator by disrupting the silenced, H-NS-bound promoter. Anti-silencing mechanisms thus control the switch between opposing lifestyles.

*For correspondence: kenneyl@uic.edu

**Competing interests:** The authors declare that no competing interests exist.

## Introduction

*Salmonella enterica* serovar Typhimurium is a rod-shaped enteric bacterium which easily infects diverse hosts such as humans, cattle, poultry and reptiles through contaminated food or water, causing gastroenteritis. A human-restricted serovar of *Salmonella enterica*, serovar Typhi, causes typhoid fever and continues to be a dangerous pathogen throughout the world. *Salmonella* lives as a facultative pathogen in various natural and artificial environments as independent planktonic cells, cooperative swarms (*Harshey and Matsuyama, 1994*) or as multi-cellular communities called biofilms (see *Steenackers et al., 2012* for a review). Upon successful invasion of host cells, *Salmonella* is phagocytosed by macrophages, where it resides in a modified vacuole in a self-nourishing niche called a Salmonella-Containing Vacuole (SCV). This intracellular lifestyle eventually adversely affects the host. *Salmonella* also resides as multi-cellular communities on intestinal epithelial cells (*Boddicker et al.,*

**eLife digest** *Salmonella* bacteria can infect a range of hosts, including humans and poultry, and cause sickness and diseases such as typhoid fever. Disease-causing *Salmonella* evolved from harmless bacteria in part by acquiring new genes from other organisms through a process called horizontal gene transfer. However, some strains of disease-causing *Salmonella* can also survive inside hosts as communities called biofilms without causing any illness to their hosts, who act as carriers of the disease and are able to pass their infection on to others.

So how do *Salmonella* bacteria 'decide' between these two lifestyles? Previous studies have uncovered a regulatory system that controls the decision in *Salmonella*, which is made up of two proteins called SsrA and SsrB. To trigger the disease-causing lifestyle, SsrA is activated and adds a phosphate group onto SsrB. This in turn causes SsrB to bind to and switch on disease-associated genes in the bacterium. However, it was less clear how the biofilm lifestyle was triggered.

Desai et al. now reveal that the phosphate-free form of SsrB – which was considered to be the inactive form of this protein – plays an important role in the formation of biofilms. Experiments involving an approach called atomic force microscopy showed that the unmodified SsrB acts to stop a major gene that controls biofilm formation from being switched off by a so-called repressor protein.

*Salmonella* acquired SsrB through horizontal gene transfer, and these findings show how this protein now acts as a molecular switch between disease-causing and biofilm-based lifestyles. SsrB protein is also involved in the decision to switch between these states, but how it does so remains a question for future work.

2002), gallstones (*Prouty et al., 2002*) and tumors (*Crull et al., 2011*). It is believed that biofilms in the gall bladder are important for maintaining the carrier state, allowing *Salmonella* to persist (*Crawford et al., 2010*).

Each of these lifestyles of *Salmonella* are regulated by two-component regulatory systems (TCRS). TCRSs are comprised of a membrane-bound sensor histidine kinase and a cytoplasmic response regulator. The virulence genes of *Salmonella* are encoded on horizontally acquired AT-rich segments of the genome called Salmonella Pathogenecity Islands (SPIs), and are also tightly regulated by TCRSs. For example, the SsrA/B TCRS is essential for the activation of the SPI-2 regulon genes encoding a type-three secretory needle and effectors that are involved in formation of the SCV (*Cirillo et al., 1998*). Interestingly, the SsrA/B system itself is regulated by upstream two-component systems such as EnvZ/OmpR and PhoP/Q, which regulate gene expression in response to changes in osmolality, pH and the presence of anti-microbial peptides (*Fields et al., 1989*; *Miller et al., 1989*; *Lee et al., 2000*; *Feng et al., 2003*). The *ssrA* and *ssrB* genes are present on the SPI-2 pathogenecity island adjacent to each other and are regulated by a set of divergent promoters (*Feng et al., 2003*; *Ochman et al., 1996*). Under acidic pH and low osmolality, the *ssrA* and *ssrB* genes are transcriptionally activated by the binding of OmpR~P and PhoP~P to their promoters (*Feng et al., 2003*; *Bijlsma and Groisman, 2005*; Walthers and Kenney unpublished) whose levels are in turn regulated by the respective sensor kinases, EnvZ and PhoQ. SsrA is a tripartite membrane-bound histidine sensor kinase that undergoes a series of intra-molecular phosphorylation reactions before it transfers the phosphoryl group to the N-terminal aspartate residue of the response regulator, SsrB.

SsrB belongs to the NarL/FixJ family of transcriptional regulators that require phosphorylation-dependent dimerization to bind DNA. The X-ray crystal structure of NarL revealed that the C-terminal DNA binding domain was occluded by the N-terminus (*Baikalov et al., 1996*), and phosphorylation was predicted to relieve this inhibition. Full-length SsrB is unstable in solution, but an isolated C-terminal domain of SsrB, SsrBc, is capable of binding to the regulatory regions of nine genes belonging to the SPI-2 regulon, including *ssrA* and *ssrB* (*Feng et al., 2004*; *Walthers et al., 2007*) and activating transcription. A role for SsrB~P was identified by its dual function as a direct transcriptional activator and as an anti-silencer of H-NS-mediated repression (*Walthers et al., 2007*). The Histone like Nucleoid Structuring protein H-NS is involved in silencing many of the SPI-2 regulon genes

in accordance with its role in binding to xenogenic AT-rich sequences and repressing their expression (*Walthers et al., 2007*; *Navarre et al., 2006*). H-NS binding to DNA leads to the formation of a stiff nucleoprotein filament which is essential in gene silencing (*Lim et al., 2012*; *Liu et al., 2010*; *Amit et al., 2003*; *Winardhi et al., 2015*). Moreover, relief of repression occurs due to the binding of SsrBc to this rigid H-NS-DNA complex (*Walthers et al., 2011*).

*Salmonella* reservoirs in host and non-host environments produce a three-dimensional extracellular matrix which consists of curli fimbriae, cellulose, proteins and extracellular DNA, to encase clusters of bacteria and form a mature biofilm. CsgD (AgfD) is the master regulator of biofilm formation (*Gerstel et al., 2003*); it is a LuxR family transcriptional activator that activates the expression of curli fimbriae encoded by *csgDEFG/csgBAC* operons (*Collinson et al., 1996*; *Romling et al., 1998*). CsgD also activates expression of *adrA*, increasing intracellular c-di-GMP levels, and activating the cellulose biosynthetic operon *bcsABZC* (*Zogaj et al., 2001*). Two other biofilm matrix components are also positively regulated by CsgD: BapA and the O-antigen capsule (*Latasa et al., 2005*; *Gibson et al., 2006*).

Transcriptional profiling of biofilms formed by *S.* Typhimurium SL1344 showed that many SPI-2 genes were down-regulated, yet SsrA was required for biofilms (*Hamilton et al., 2009*). This apparent paradox drove us to explore the underlying mechanism of biofilm formation. The role of SsrA/B in this process was of particular interest, since our previous comparison of SsrA and SsrB levels at neutral and acidic pH had shown that the expression of *ssrA* and *ssrB* was uncoupled (*Feng et al., 2004*).

We examined the ability of the wild type *S.* Typhimurium strain 14028s to form biofilms in the absence of *ssrA* and *ssrB* and found it to be dependent only on the expression of *ssrB*. We further showed that H-NS was a negative regulator of *csgD.* Surprisingly, the SsrB response regulator positively regulated the formation of biofilms by activating *csgD* expression in the absence of any phospho-donors. Moreover, AFM imaging revealed that unphosphorylated SsrB was able to bind to the *csgD* regulatory region and binding was sufficient to relieve H-NS-mediated repression and favor formation of *S.* Typhimurium biofilms.

As a result of these studies, we propose that SsrB, a pathogenicity island-2-encoded response regulator, sits at a pivotal position in governing *Salmonella* lifestyle fate: to either exist inside the host (in the SCV) as a promoter of virulence; or as a surface-attached multicellular biofilm, maintaining the carrier state. This switch is achieved merely by the ability of unphosphorylated SsrB to function as an anti-repressor of H-NS and the additional role of SsrB~P in activating SPI-2 transcription (*Walthers et al., 2011*).

## Results

### SsrB supports biofilm formation in the absence of its kinase SsrA

The SsrA/B TCRS is activated by environmental stimuli such as pH and osmolality, stimulating the expression of virulence genes essential for intra-cellular growth and survival of *Salmonella* (*Feng et al., 2003*; *2004*). Thus, it was surprising that SsrA/B and SPI-2 were implicated in the multi-cellular lifestyle of *Salmonella* (*Hamilton et al., 2009*). Furthermore, the SsrA kinase was required, but SPI-2 genes were down-regulated. At SPI-2, SsrB~P de-represses H-NS and also activates transcription (*Walthers et al., 2007*; *2011*). We therefore wanted to explore this seeming paradox during biofilm regulation.

In order to quantify the defect in biofilm-forming ability resulting from the absence of *ssrB*, the wild type strain 14028s, an *ssrA* null strain and an *ssrB* null strain were grown in 96-well polystyrene microtiter plates for 2 days. The surface-attached cells in 14028s, *ssrA* and *ssrB* strains were stained by crystal-violet and quantified. As shown in the figure, biofilm levels in the *ssrB* mutant strain decreased by around 60% compared to the wild type or an *ssrA* null strain (*Figure 1A*), and it could be complemented by over-expressing SsrBc *in trans*. Thus, we establish a new role for SsrB, but not the SsrA kinase in positively regulating biofilm formation. The *ssrB* strain was not compromised for planktonic growth, as measured by total viable counts of the wild type, *ssrA* and *ssrB* cultures (*Figure 1—figure supplement 1* and Supplementary methods).

The ability to form biofilms was analysed for three strains, wild type 14028s, an *ssrA* null strain and an *ssrB* null strain, by observing appearance of the rough dry and red (rdar) morphotype on

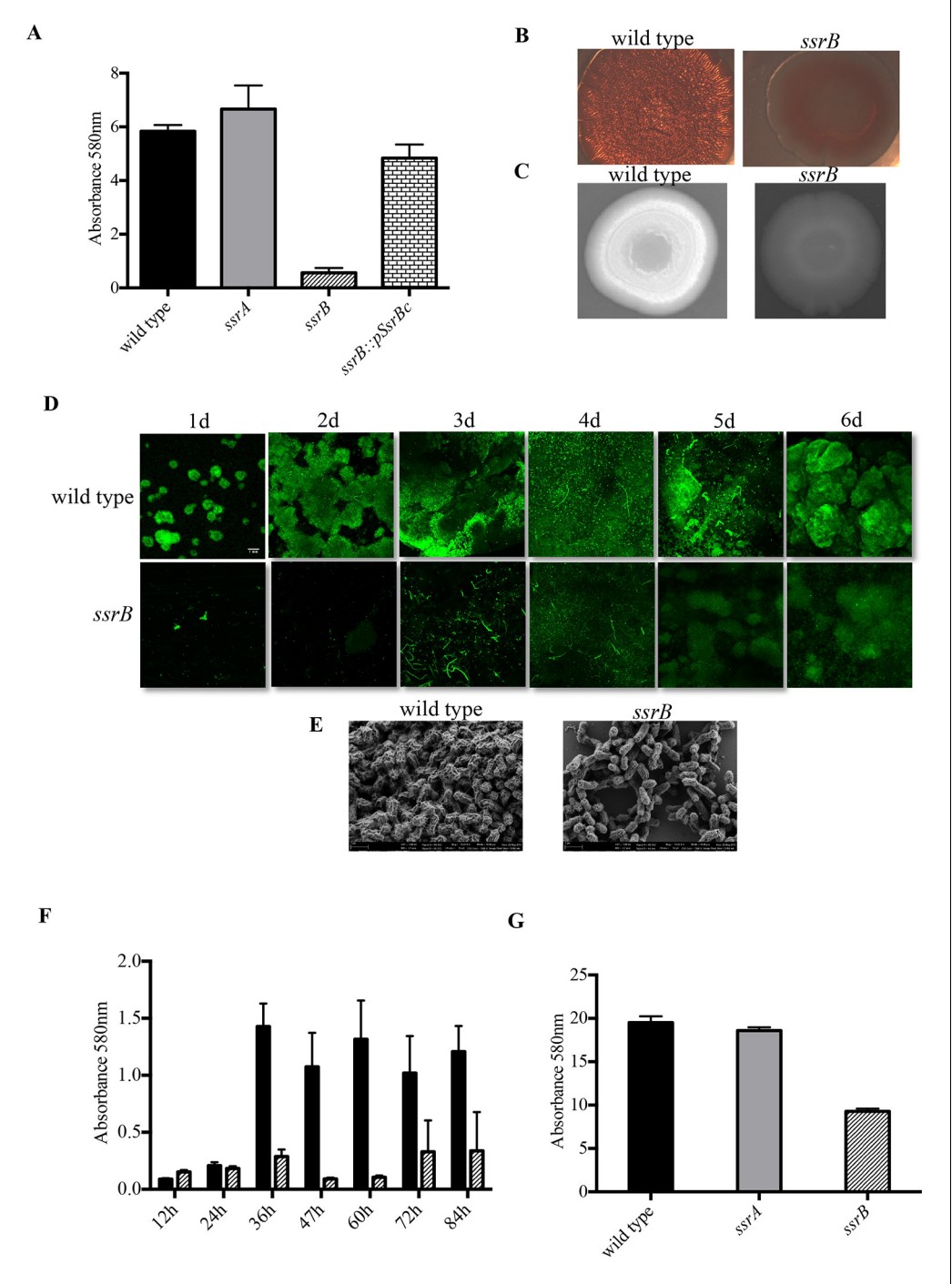

**Figure 1.** Loss of *ssrB* but not *ssrA* decreases *Salmonella* Typhimuirum biofilms. (**A**) The defect in formation of biofilms in the *ssrB* null was complemented by the overexpression of SsrBc from plasmid pKF104 *in trans* as measured by crystal violet staining. (**B**) The typical rdar morphotype of the wild type strain was lost in the *ssrB* strain as shown on congo red plates. (**C**) A two day old macrocolony of the *ssrB* strain is not fluorescent under UV light with Fluorescent Brightener 28. (**D**) The wild type strain forms thick solid-surface biofilms, while the *ssrB* strain remains poor for biofilms as monitored for six days by SYTO-9 staining of flow cell biofilms; scale bar = 1 mm. (**E**) SEM images showing extensive mesh-like network of wild type biofilms and sparse extracellular matrix of the *ssrB* biofilms; scale bar = 1 μm. (**F**) The amount of biofilms formed by the wild type strain (solid black bars) increases after 24 hr but the *ssrB* null (hatched black bars) remains defective up to 84 hr. n = 2, Mean ± SD, p < 0.0001 between wild type and *ssrB* strains from 36 hr till 84 hr. (**G**) The amount of cholesterol-attached biofilms formed by

*Figure 1 continued on next page*

*Figure 1 continued*

the *ssrB* strain were significantly less than that produced by the wild type. n = 3, Mean ± SD, p < 0.0001. Source data file: *Figure 1—source data 1*.

The following source data and figure supplements are available for figure 1:

**Source data 1.** Source data for crystal violet staining in *Figure 1A,F and G*.

**Figure supplement 1.** The *ssrB* mutant is not defective in growth compared to the wild type strain.

**Figure supplement 1—source data 1.** Growth curves of wild type, *ssrA, ssrB* and D56A strains.

**Figure supplement 2.** The planktonic sub-population of the *ssrB* strain was higher by two orders of magnitude compared to the wild type, *ssrA* and *D56A* strains at 2 days.

**Figure supplement 2—source data 1.** Number of cells in the planktonic sub-population of each strain.

**Figure supplement 3.** Total wet weight of the adherent sub-population was decreased by at least 50% in the *ssrB* strain compared to the wild type, *ssrA* and D56A strains at 2 days.

**Figure supplement 3—source data 1.** Total wet weight of the adherent sub-population of each strain.

Congo Red plates. After seven days, each of the 14028s and *ssrA* mutant plates showed the appearance of rough, dry and red macrocolonies. However, the *ssrB* null strain exhibited a smooth, wet and pale brown morphotype which showed a poor ability to exhibit a multicellular phenotype (*Figure 1B*). The absence of Congo Red staining in the *ssrB* strain also indicated that there were greatly reduced levels of extracellular curli fibers in the macrocolony. These fibers form one of the key components of the extra-cellular matrix of a *Salmonella* biofilm. To examine the levels of cellulose, the other main component of the extra-cellular matrix, the three strains were grown on Calcofluor plates. The macrocolonies formed by 14028s and *ssrA* appeared white under UV, as cellulose binds the Calcofluor fluorescent dye (*Figure 1C*). The dull and non-white macrocolony produced by the *ssrB* null strain was starkly different, indicating again a poor extracellular matrix.

We visualized the nature of biofilms formed by the wild type and *ssrB* strains by Confocal Fluorescence Imaging of solid-surface biofilms formed under a continuous flow system in a micro-fluidic chamber. Wild type and the *ssrB* null strain were grown in minimal M9 medium and inoculated in their respective flow cells. After allowing the cells to attach for 1h, flow commenced and the growing biofilms were monitored daily for six days. Each day, the flow cell was removed, and the glass-attached cells were stained with SYTO-9 green, a fluorescent dye that stains nucleic acids. The cells were then imaged using fluorescence microscopy (*Figure 1D*). The figure clearly shows that for the wild type strain, the attached biomass not only increased progressively in size, but also in its complexity of design. This correlated with the appearance of typical mushroom-like constituent microcolonies by around 2 days. Such type of mature biofilms were never observed in the *ssrB* null strain and even after 6 days it failed to form biofilms, as evident by fewer and thinly attached clusters of SYTO-9 green-labeled cells.

To compare the differences in the structural properties of biofilms formed by the *ssrB* strain with the wild type and the *ssrA* mutant, cells were grown on APTES-coated cover slips in 16-well polystyrene microtiter plates for two days. The surface-attached communities were fixed, dehydrated and dried for Scanning Electron Microscopy (SEM). As expected, the wild type and *ssrA* mutant showed the presence of tightly knit communities of cells containing an extensive network of curli fibres surrounding large groups of cells (*Figure 1E*). In contrast, the *ssrB* null strain formed sparsely spaced groups of few cells with a large reduction in the surrounding matrix and poor network properties.

It was possible that the *ssrB* strain was just slower in biofilm formation and needed more time to exhibit its biofilm capability or that it caused a growth defect. We ruled out this possibility by measuring growth curves (*Figure 1—figure supplement 1*) and by monitoring the wild type and *ssrB* strains by crystal-violet staining every 12 hr for around 4 days. This determined that the *ssrB* mutant

was not deficient for growth and remained incapable of forming biofilms up to 84 h, while the wild type formed large biofilms by 2 days (*Figure 1F*).

We next investigated the ability of *Salmonella* to form biofilms, on cholesterol-coated eppendorf tubes. The cholesterol-attached biofilms were estimated by crystal violet staining for the strains 14028s, *ssrA* null and *ssrB* null. There was a drastic reduction in the amount of biofilms formed by the *ssrB* strain on cholesterol-coated eppendorf tubes (*Figure 1G*). If cholesterol biofilms are indeed an indicator of an ability to form biofilms on gallstones (*Crawford et al., 2008*), then SsrB may be crucial for establishment of the carrier state in *Salmonella*.

Taken together, our results modify the initial hypothesis that SsrA was involved in the formation of biofilms. SsrB, but not its cognate kinase SsrA, is involved in the switch from planktonic growth to a multi-cellular lifestyle, in addition to its role in regulating genes required for pathogenesis.

## Phosporylation of SsrB is not required for biofilm formation

Phospho-relay between the sensor kinase, SsrA, and the response regulator, SsrB, was crucial for activation of SPI-2 virulence genes (*Feng et al., 2003*). SsrA is a tripartite sensor kinase that is presumably autophosphorylated by ATP at His405, followed by intra-molecular phosphorylation reactions at Asp739 and His867, based on homology to other tripartite kinases. His867 would then participate in the transfer reaction of the phosphoryl group to Asp56 in the N-terminus of SsrB. To confirm that SsrA-dependent phosphorylation was not required for SsrB-mediated regulation of biofilms (*Figure 1A–G*), we examined biofilm formation of *Salmonella* strains possessing H405Q, D739A and H867Q mutations in *ssrA* (Walthers and Kenney unpublished) and compared them to the wild type, *ssrA* and *ssrB* null mutants. Figure 3A indicates that the extent of biofilms formed by the three kinase point mutants was comparable to that of the wild type and the *ssrA* null mutant. A complete loss of *ssrB* was the only genetic change which adversely affected biofilm formation.

In the absence of SsrA kinase activity, small inorganic phosphate donors such as acetyl phosphate can phosphorylate SsrB, albeit with a lower efficiency (*Feng et al., 2004*). We therefore constructed strains that were deficient in both SsrA kinase activity as well as in the production of acetyl phosphate and assayed their biofilm capabilities. Strains H405Q *ack pta*, D739A *ack pta* and H867Q *ack pta*, were all capable of biofilm formation (*Figure 2A*), demonstrating that none of the known phosphate donors for SsrB were required for regulating biofilms. In contrast, SPI-2 activation required *ssrA*, as measurement of β-galactosidase activity of a *sifA-lacZ* transcriptional fusion confirmed that activation of *sifA* required SsrA/B phosphorylation (*Figure 2C*).

SsrB is in the NarL subfamily of response regulators. NarL requires phosphorylation to relieve inhibition of C-terminal DNA binding by the N-terminus (*Baikalov et al., 1996*). Therefore, we substituted the conserved Asp56 residue with Ala (D56A), and determined its ability to form biofilms (*Figure 2B*). The D56A strain was as competent as the wild type, *ssrA* null and *ssrA* kinase mutants in biofilm formation. The ability of unphosphorylated SsrB to regulate biofilm formation was unique, as it was unable to activate *sifA-lacZ* levels. The beta-galactosidase activity was significantly lower in the *ssrB* null and D56A strains (8.7% and 14.9%, respectively) compared to the wild type (*Figure 2C*). The D56A mutant was also able to form rdar macrocolonies on Congo Red plates. In addition, SEM imaging of D56A biofilms revealed that they maintained their intricate architecture and wild type structure (data not shown).

When cultures of the wild type, *ssrA, ssrB* and D56A strains were separated as a non-adherent (or free-swimming) sub-population and an adherent sub-population (or multicellular aggregates) (*MacKenzie et al., 2015*), we observed that the *ssrB* strain had a greater number of cells in the non-adherent fraction and fewer in the adherent fraction as compared to the other three strains. This confirms that the absence of SsrB, and not SsrB~P, led to a defect in the ability of *Salmonella* to form surface-attached communities (*Figure 1—figure supplement 2* and *3*).

Taken together, these results indicate that SsrB can activate biofilm formation in the absence of any phosphorylation signals. This is the first evidence for a role of unphosphorylated SsrB in gene regulation. These results also strongly advocate for a larger contribution of SsrB in *Salmonella* pathogenesis owing to its dual regulation of intracellular (i.e., the SCV) as well as mutli-cellular lifestyles (the carrier state).

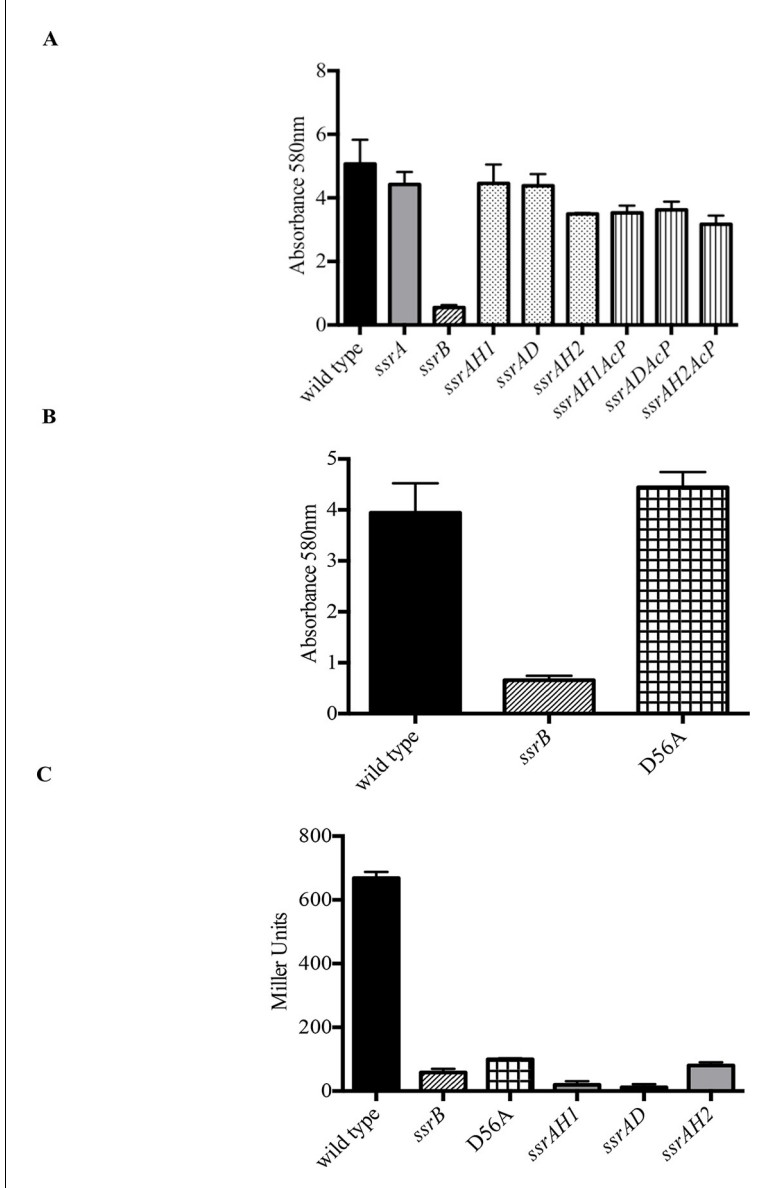

**Figure 2.** Phosphorylation of SsrB is not required for biofilm formation. Amount of biofilms formed as measured by crystal violet staining for (**A**) Strains *ssrAH1, ssrAD, ssrAH2, ssrAH1AcP, ssrADAcP, ssrAH2AcP* and (**B**) D56A SsrB shows similar levels to that of the wild type, and higher than the *ssrB* mutant. Source data file: *Figure 2—source data 1*. (**C**) Beta-galactosidase activity of a *sifA-lacZ* chromosomal fusion was significantly lower in the *ssrB* null and the D56A SsrB mutant compared to the wild type. n = 3, Mean ± SD, p < 0.0001. Source data file: *Figure 2—source data 2*.

The following source data is available for figure 2:

**Source data 1.** Source data for crystal violet staining in *Figure 2A and B*.

**Source data 2.** Source data for the measurement of beta-galactosidase activity in *Figure 2C*.

## SsrB-dependent biofilm formation does not require SPI-2

SsrB is encoded on SPI-2 and regulates SPI-2 genes involved in assembly and function of the type three secretion system encoded by SPI-2, as well as effectors that are encoded on SPI-2 and outside of SPI-2 (*Feng et al., 2004*; *Walthers et al., 2007*). To determine whether the SsrB target of biofilm

regulation was dependent on the SPI-2 injectisome or any of its secreted SPI-2 effectors, we examined *ssaC* and *ssaJ* null strains for their ability to form biofilms (*Figure 3A*). SsaC is an outer ring component of the injectisome and SsaJ forms the inner ring of the SPI-2 needles. Both of these strains formed biofilms to an extent similar to the wild type and hence, we ruled out any involvement of the SPI-2 secretory apparatus or its related secreted proteins in the SsrB-dependent regulation of *Salmonella* biofilms.

## Unphosphorylated SsrB activates *csgD* expression

Biofilm formation in *Escherichia coli* and *Salmonella* Typhimurium is governed by the master regulator, CsgD, that acts as a transcriptional activator of genes involved in curli biogenesis and cellulose synthesis. Some of the environmental conditions that favor the formation of biofilms in *Salmonella* such as low salt and acidic pH also up-regulate levels of SsrB in the cell (*Feng et al., 2003*). An obvious null hypothesis was therefore to test whether SsrB (in its unphosphorylated state), activated the expression of *csgD*. We first over-expressed *csgD* from a plasmid in the *ssrB* null mutant and determined whether it could rescue the defect in biofilm formation. The presence of *csgD* in *trans* restored the biofilm capability to wild type levels as measured by crystal-violet staining of 2 day old biofilms (*Figure 3B*).

We next examined the levels of *csgD* transcripts in 2 day macrocolonies formed by the wild type, *ssrA* null, *ssrB* null and D56A strains by quantitative real time RT-PCR (qRT-PCR). As shown in *Figure 3C*, there was around a 60-fold decrease in *csgD* transcripts when *ssrB* was deleted (*Figure 3C*). We also corroborated our previous findings (*Figure 1A* and *2B*), as *csgD* transcripts in the *ssrA* null and D56A strains were maintained at levels similar to the wild type. Furthermore, we probed the whole-cell lysates obtained from such macrocolonies with a monoclonal anti-CsgD antibody in order to measure the CsgD protein levels by western blotting. CsgD levels were undetected in the *ssrB* null compared to the wild type, *ssrA* null or D56A strains (*Figure 3D*). Hence, unphosphorylated SsrB was able to up-regulate the expression of *csgD* and positively influence the shift to a sessile lifestyle.

## SsrB and H-NS differentially regulate *csgD* expression

Both in *E. coli* and *S.* Typhimurium, the expression of *csgD* is sensitive to various environmental stimuli such as starvation, temperature, pH and osmolality due to the action of upstream global regulators at the intergenic region of the *csgDEFG* and *csgBAC* operons. We noted that the SsrA/B system also responded to similar environmental conditions such as acidic pH and low osmolality. Furthermore, H-NS, a known repressor of SPI-2 genes (*Walthers et al., 2007*), was also known to regulate the expression of *csgD* in *E. coli* and *S.* Typhimurium (*Ogasawara et al., 2010*; *Gerstel et al., 2003*). In regulating SPI-2 genes, SsrBc and H-NS work in an opposing fashion (*Walthers et al., 2011*). SsrBc antagonises the repressive action of H-NS at regulatory regions upstream of the promoters, while also acting as a direct transcriptional activator. We therefore examined whether or not this paradigm for SsrB-mediated transcriptional activation was applicable to *csgD* regulation.

In order to test this, we completely deleted *hns* from the *ssrB* null strain and determined if its deletion rescued biofilm formation. Crystal-violet staining of static biofilms formed by the wild type, *ssrB* and *ssrB hns* strains was performed; the amount of biofilms formed by *ssrB hns* was the same as the wild type. In contrast, the *ssrB* null strain remained at around 35% of these levels (*Figure 4A*). Macrocolonies formed by the *ssrB hns* strain also displayed a rugose morphology, reminiscent of the wild type strain after 2 days (*Figure 4B*). In addition, SEM images of 2 day-old biofilms formed by the *ssrB hns* strain showed a typical 'biofilm' architecture, i.e., densely packed communites of cells surrounded by biofilm matrix (*Figure 4C*). This result indicated that H-NS was functioning to repress *csgD* (see Discussion). Thus, we strengthened our prediction of a role for SsrB in the activation of *csgD* expression by acting as an anti-H-NS molecule.

We next examined the levels of *csgD* transcripts in mature biofilms formed by the wild type, *ssrB* null, *hns* null and *ssrB hns* null strains by qRT-PCR (*Figure 4D*). Normalized levels of *csgD* transcripts were highly up-regulated in the *ssrB hns* double mutant compared to the *ssrB* null strain, indicating that the loss of H-NS repression rescued the defect in *csgD* expression in the absence of SsrB. Expression of *csgD* was slightly higher than the wild type levels in the *hns* single mutant and the *ssrB hns* double mutant, which also correlated with their biofilm capabilities (*Figure 4A*). Since the levels

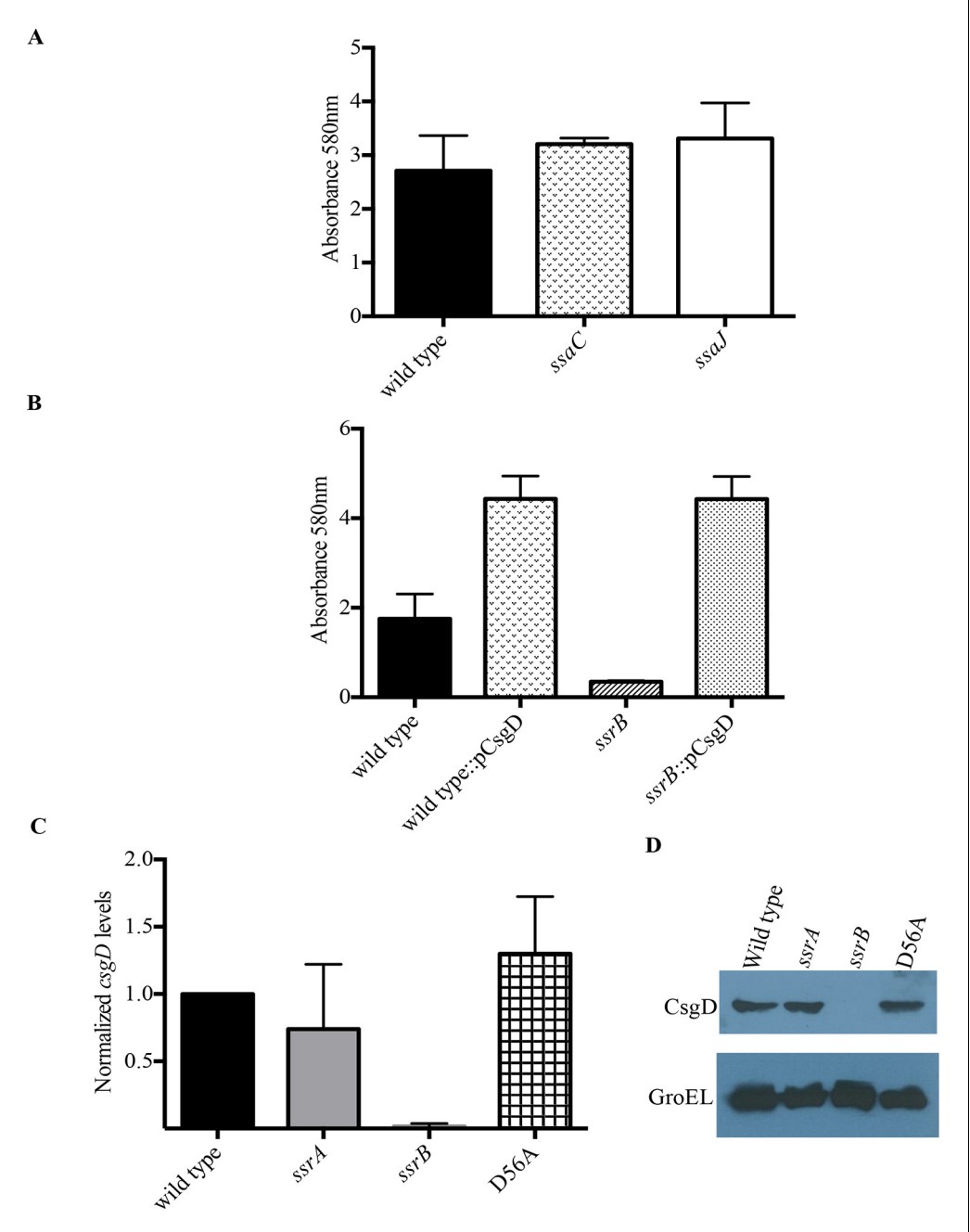

**Figure 3.** SsrB regulates biofilms by a CsgD-dependent mechanism. (**A**) The SPI-2 needle, *ssaC* and *ssaJ* mutant strains were not affected in biofilm formation. (**B**) Over-expression of *csgD* from a plasmid (pBR328::csgD) *in trans* rescued biofilm formation in the *ssrB* mutant, as measured by crystal violet staining, n = 3. Source data file: *Figure 3—source data 1*. An estimate of *csgD* expression by (**C**) Real-time qRT-PCR showed a significant decrease in *csgD* transcription in the *ssrB* null, but not in the D56A SsrB and *ssrA* mutants. *rrsA* transcript levels were used as control; n = 2, Mean ± SD, p < 0.0001. Source data file: *Figure 3—source data 2* and (**D**) Immunoblot analysis showing the absence of CsgD in the *ssrB* null strain in two day old biofilms, using GroEL as a loading control.

The following source data is available for figure 3:

**Source data 1.** Source data for crystal violet staining in *Figure 3A and B*.
**Source data 2.** Source data for Real-time qRT-PCR in *Figure 3C*.

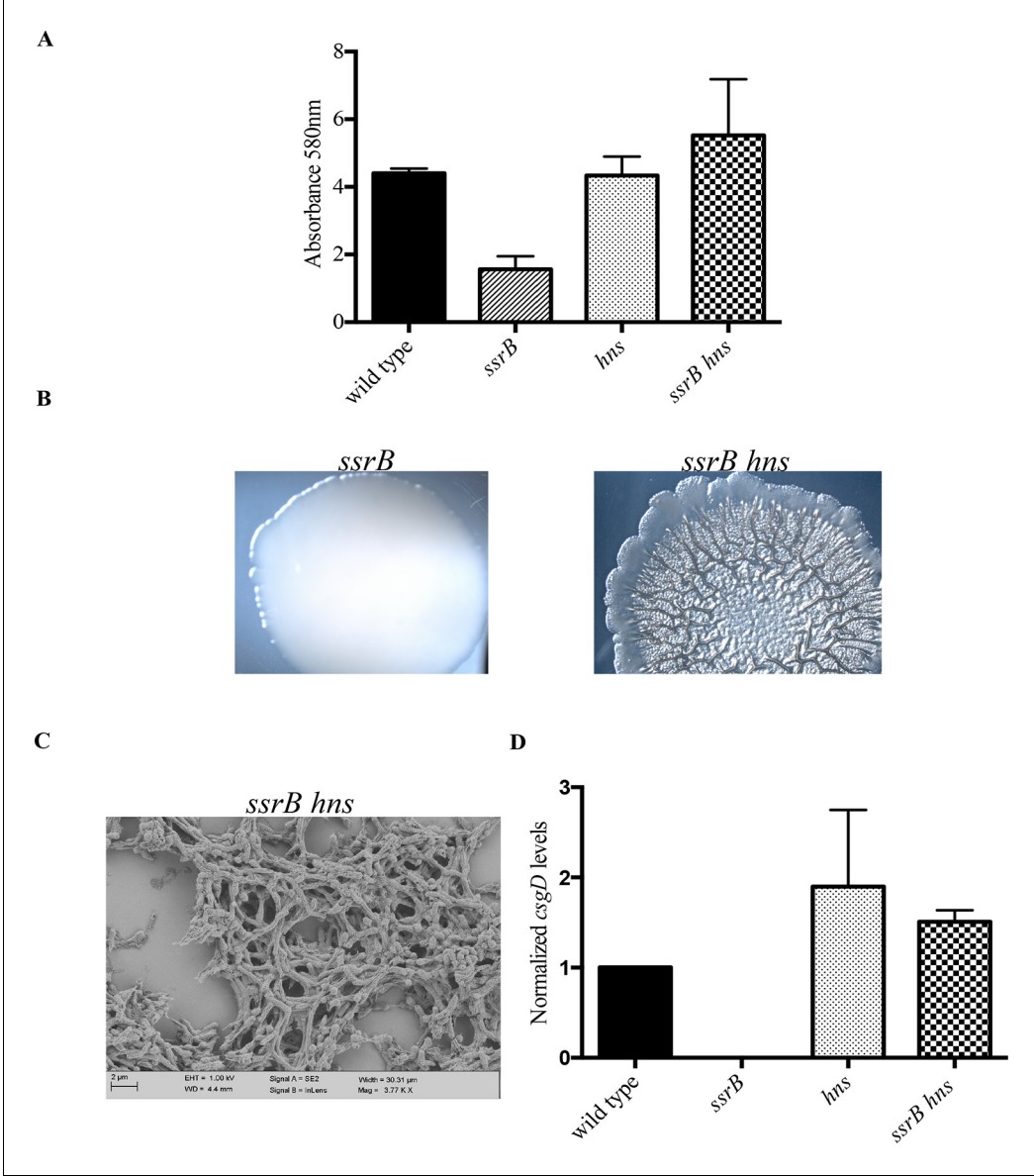

**Figure 4.** *hns* deletion rescues biofilm formation in the *ssrB* mutant as shown by Crystal violet staining. (**A**) The amount of biofilms formed is higher in the wild type, *hns* and *ssrB hns* strains compared to the *ssrB* null, n = 3, Mean ± SD, p < 0.0001. Source data file: *Figure 4—source data 1*. Macrocolony phenotype (**B**) *ssrB hns* forms a highly rugose and dry macrocolony, while the *ssrB* macrocolony was smooth and mucoidy. SEM imaging (**C**) *ssrB hns* biofilms were covered by a thick extra-cellular matrix; scale bar = 2 μm. (**D**) qRT-PCR: *csgD* levels were restored in the *ssrB hns* strain and were higher than the wild type (p < 0.03) and the *ssrB* mutant (p < 0.003) against *rrsA* transcripts as a control. Note that the normalized *csgD* levels in the *ssrB* null were 0.0009, too low for the scale. n = 2, Mean ± SD. Source data file: Source data file: *Figure 4—source data 2*.

The following source data is available for figure 4:

**Source data 1.** Source data for crystal violet staining in *Figure 4A*.
**Source data 2.** Source data for Real-time qRT-PCR in *Figure 4D*.

of *csgD* transcripts were equal in the *hns* and *ssrB hns* null strains, it seemed likely that SsrB influenced *csgD* expression by silencing H-NS-mediated repression and not by direct activation of transcription, which requires phosphorylation for a productive interaction with RNA polymerase (*Walthers et al., 2011*).

Taken together, these results identified a regulatory role for unphosphorylated SsrB by orchestrating anti-silencing at the H-NS-repressed *csgD* locus. This work clearly shows that, as observed in *E. coli*, H-NS represses expression of *csgD* in *Salmonella* (see Discussion).

## Unphosphorylated SsrB binds to the *csgD* regulatory region

SsrB is a NarL family member and X-ray crystallography suggested that phosphorylation was required for DNA binding (*Baikalov et al., 1996*). Thus, it was of interest to determine how SsrB relieved H-NS silencing at *csgD*. We used atomic force microscopy (AFM) to visualize the *csgDEFG-csgBAC* intergenic region and examined the effect of addition of full-length SsrB, D56A SsrB and SsrBc (DNA binding domain alone). Surprisingly, we observed binding of SsrB, D56A SsrB and SsrBc to distinct regions of the *csgD* regulatory region (*Figure 5B* and *Figure 5—figure supplement 1A and B*). Binding occured at low protein concentrations, suggesting high affinity interactions and showed that unphosphorylated SsrB was capable of binding *csgD*. An SsrB mutant, K179A, that was incapable of binding DNA (*Carroll et al., 2009*) did not bind *csgD* even at 300 nM, indicting that binding was specific (*Figure 5E and F*). Closer examination revealed a sharp curvature at the regions where SsrB was bound to *csgD* (*Figure 5B*, arrows). This result indicates that, like its NarL homologue (*Maris et al., 2002*), SsrB bends DNA upon binding, as predicted (*Carroll et al., 2009*). We estimate that on average, binding of SsrB to the *csgDEFG-csgBAC* intergenic sequence occurred with a bending angle of around 82° (*Figure 5D* and *Figure 5—figure supplement 2* for naked DNA). The observation that SsrB bending was more severe than NarL (82° compared to the 42° of NarLc at its cognate palindrome) (*Maris et al., 2002*), likely contributes to its ability to promote anti-silencing.

When the SsrB concentration was increased to 300 nM, we detected large-scale condensation of the DNA-protein complexes by AFM (*Figure 5C*). Condensation at the *csgD* regulatory region was similar when incubated with 300 nM D56A SsrB or SsrBc, indicating that protein binding led to DNA structural changes, irrespective of whether SsrB was phosphorylated (*Figure 5—figure supplement 1C and D*). Biochemical analysis of the protein-DNA complexes by an electrophoretic mobility shift assay using a shorter fagment of the *csgD* regulatory region (*Figure 6—figure supplement 4*), also indicated the presence of an SsrB-DNA complex. This complex was dissociated when an unlablelled *csgD* fragment was added as a competitor. In contrast, K179A SsrB failed to form a complex with DNA (*Figure 5—figure supplement 3*). The DNA bending ability of SsrB is important for SsrB relief of H-NS-mediated transcriptional silencing at *csgD* (see below and [*Winardhi et al., 2015*]). The DNA binding behavior of SsrB was recapitulated in solution AFM imaging experiments, indicating that binding and condensation were not an artifact of drying the samples (*Figure 6—figure supplement 1C*).

## SsrB binds an H-NS stiffened nucleoprotein filament at *csgD*

Our previous work established that H-NS silenced genes by forming a rigid filament on DNA (*Liu et al., 2010*; *Walthers et al., 2011*; *Lim et al., 2012*). We purified H-NS, incubated it with the *csgD* regulatory region, and immobilised it on a glass coverslip. Subsequent AFM imaging indicated the presence of a straight and rigid nucleoprotein filament (*Figure 6A(i)*), which was distinct from the random conformation adopted by naked DNA (*Figure 5A*). Thus, we reaffirmed our earlier observations that H-NS repressed expression of *csgD* by filament formation, leading to transcriptional silencing (*Figure 4D*).

When we pre-formed stiffened filaments by addition of H-NS and then added SsrB, evidence of SsrB condensation was immediately apparent (*Figure 6A(ii)*). For example, areas of SsrB binding to DNA led to the formation of condensed nucleoprotein complexes and abolished the prior structural rigidity introduced by H-NS binding (see arrows, *Figure 6A(ii)*). Interestingly, *Figure 6A(ii)* also indicates that H-NS continued to form straight nucleoprotein complexes in regions devoid of SsrB (yellow line), i.e. H-NS was still bound to some regions of DNA when SsrB was also bound. Similar binding behavior with D56A SsrB and SsrBc at the H-NS-bound *csgD* regulatory region was also

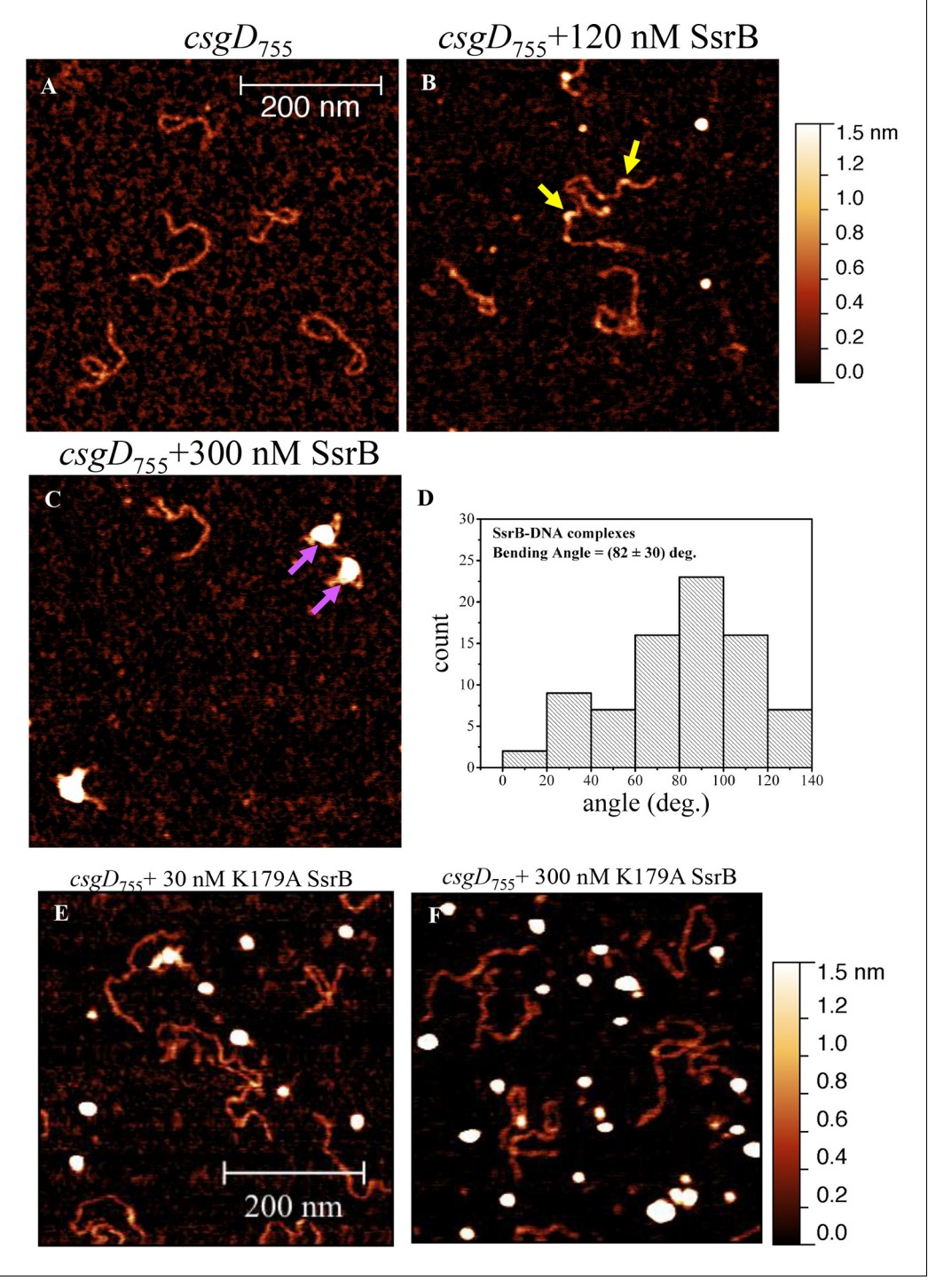

**Figure 5.** SsrB binds upstream of *csgD*. (**A**) AFM images of the 755 bp *csgD* regulatory region (*csgD₇₅₅*). (**B**) At 120 nM SsrB, distinct areas of SsrB binding were visualized as sharp bends (yellow arrows). (**C**) At 300 nM SsrB, areas of condensation (pink arrows) were observed. (**D**) Binding of SsrB bends the DNA by an average angle of 82° (for the naked DNA angle, refer to *Figure 5—figure supplement 2* and for analysis refer to Supplementary method), Scale bar = 200 nm as in (**A**). (**E**) and (**F**) The SsrB mutant, K179A SsrB, which is defective in DNA binding, was unable to bind *csgD₇₅₅* both at 30 nM or 300 nM; Scale bar = 200 nm.

The following figure supplements are available for figure 5:

*Figure 5 continued on next page*

*Figure 5 continued*

**Figure supplement 1.** AFM images of the 755 bp *csgD* regulatory region (*csgD$_{755}$*) (**A**) with 120 nM D56ASsrB and (**B**) SsrBc.

**Figure supplement 2.** Bending angle of the naked *csgD$_{755}$* fragment.

**Figure supplement 3.** Electrophoretic mobility shift assay with the 122 bp *csgD* regulatory region, *csgD$_{122}$*, showing a DNA-protein complex in the presence of SsrB.

observed (*Figure 6—figure supplement 2A and B*), and similar binding patterns were observed in solution AFM (*Figure 6—figure supplement 1B–D*). This combined nucleoprotein complex was also detected as a supershift by electrophoretic mobility shift assay using a shorter fragment of the *csgD* regulatory region and specific concenterations of H-NS, SsrB and anti-SsrBc serum (*Figure 6—figure supplement 3*). This *csgD* regulatory element harbored the H-NS binding region (*Gerstel et al., 2003*) as well as an SsrB binding motif (*Feng et al., 2004*; refer *Figure 6—figure supplement 4*). Thus, unphosphorylated SsrB can bind to the *csgD* regulatory region when it has been coated with the repressor H-NS. Anti-silencing results from SsrB-induced local topological changes in the DNA, in part as a result of its bending ability (*Figure 6B*). This likely provides enough free DNA to enable access to the promoter for RNA polymerase to activate transcription.

## Discussion

Pathogenic microbes constantly evolve novel means to counter the multitude of challenges posed by complex eukaryotic hosts. Successful acquisition and integeration of laterally acquired genes into the native genome of pathogens leads to novel capabilities enabling their survival in a wide range of environmental stresses. The present work demonstrates how the presence or absence of the horizontally acquired SsrA kinase controls post-translational modification of the transcription factor SsrB (i.e. phosphorylation at aspartate-56). This event controls the fate of *Salmonella* Typhimurium, resulting in either acute or chronic, but asymptomatic infection. A variation on two-component signaling in a similar lifestyle fate in *Pseudomonas aeruginosa* involved the presence or absence of the hybrid kinase RetS (*Goodman et al., 2004*).

### SsrB sits at a pivotal decision point that determines *Salmonella* lifestyles

When the SsrA kinase is present and activated by acid stress, SsrB is phosphorylated and SsrB~P de-represses H-NS and activates transcription at SPI-2 and SPI-2 co-regulated genes, including: *sifA* (*Walthers et al., 2011*), *ssaB*, *ssaM*, *sseA* and *ssaG* (*Walthers et al., 2007*). In the absence of the SsrA kinase, SsrB is not phosphorylated, but it can counter H-NS silencing at *csgD* (*Figure 4A–D* and *Figure 6A*). SsrB binding and bending at the *csgD* promoter causes a sufficient change in the DNA secondary structure (*Figure 5B,C*) that likely enables access for RNA polymerase, stimulating *csgD* transcription. It is interesting to note that SsrB is located on the SPI-2 pathogenicity island, and thus was acquired as *Salmonella enterica* diverged from *Salmonella bongori*. However, the capability to form biofilms is an ancestral trait, as phylogeny studies have shown that most of the natural or clinical isolates of *Salmonella* belonging to all the five sub-groups form rdar colonies (*White and Surette, 2006*). The SsrB response regulator can control two distinct lifestyle choices: the ability to assemble a type three secretory system and survive in the macrophage vacuole or the ability to form biofilms on gallstones in the gall bladder to establish the carrier state.

   What then controls the presence or activation of the kinase SsrA? Our early experiments indicated that SsrA and SsrB were uncoupled from one another (i.e., SsrB was present in the absence of SsrA) and *ssrA* transcription was completely dependent on OmpR (*Feng et al., 2004*). The EnvZ/OmpR system is stimulated by a decrease in cytoplasmic pH when *Salmonella* enters the macrophage vacuole (*Chakraborty et al., 2015*). This may also be the stimulus for activating SsrA, since the *Salmonella* cytoplasm acidifies to pH 5.6 during infection and the cytoplasmic domain of EnvZ (EnvZc) was sufficient for signal transduction (*Wang et al., 2012*; *Chakraborty et al., 2015*).

Previous reports also identified a role for PhoP in *ssrA* translation (*Bijlsma and Groisman, 2005*), which would further add to fluctuating SsrA levels. The present work describes a novel role for the unphosphorylated response regulator SsrB in de-repressing H-NS (*Figure 6B*). We show that under biofilm-inducing conditions, unphosphorylated SsrB is sufficient to activate the expression of *csgD*. There are only a few such examples of unphosphorylated response regulators playing a role in

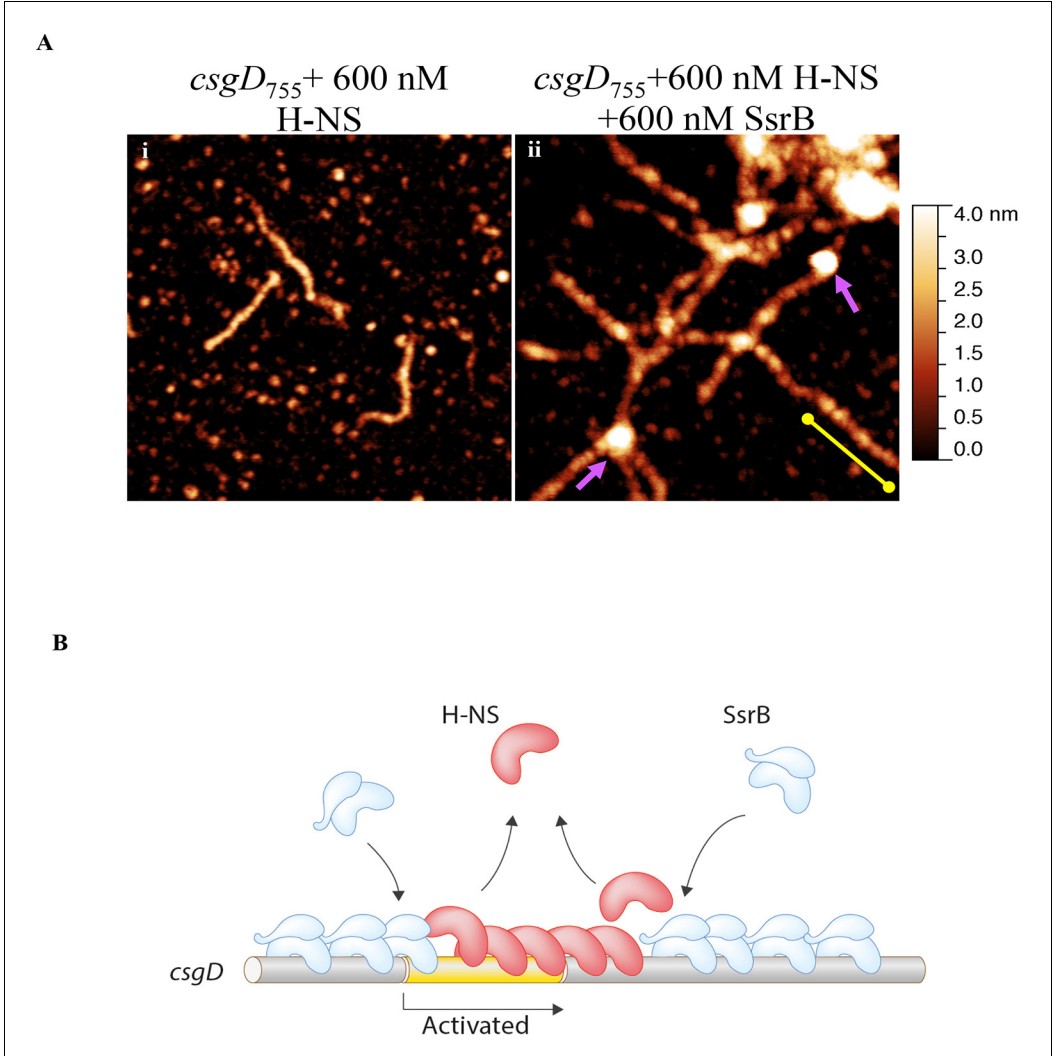

**Figure 6.** SsrB condenses H-NS bound *csgD* DNA. (**A**) (i) AFM imaging in the presence of 600 nM H-NS shows a straight and rigid filament on *csgD*$_{755}$. (ii) Addition of 600 nM SsrB to the H-NS bound *csgD* DNA resulted in areas of condensation (pink arrows; an 'SsrB signature') along with a few areas where the straight H-NS bound conformation persisted (yellow line; an 'H-NS signature'); Scale bar = 200 nm as in *Figure 5A*. (**B**) A model for the mechanism of anti-silencing by SsrB at *csgD* wherein SsrB likely displaces H-NS from the ends of a stiffened nucleoprotein filament and relieves the blockade on the promoter for RNA polymerase to activate transcription. For details refer to (*Winardhi et al., 2015*).

The following figure supplements are available for figure 6:

**Figure supplement 1.** Liquid AFM imaging of (**A**) the 755 bp *csgD* regulatory region.

**Figure supplement 2.** SsrB D56A and SsrBc condense H-NS-bound *csgD* DNA.

**Figure supplement 3.** SsrB and H-NS form a complex on *csgD*.

**Figure supplement 4.** The sequence of the 755 bp *csgD* regulatory region indicating the H-NS binding region according to *Gerstel et al. (2003)*; and the SsrB binding motif as found by *Feng et al. (2004)*.

transcription such as DegU (*Dahl et al., 1992*) in *Bacillus subtilis* and RcsB (*Latasa et al., 2012*) in *S.* Typhimurium.

## The importance of anti-silencing in gene regulation

In recent years, it has become apparent that H-NS silences pathogenicity island genes in *Salmonella* (*Lucchini et al., 2006*; *Navarre et al., 2006*; *Walthers et al., 2007*; *2011*). Understanding how H-NS silences genes and how this silencing is relieved is an active area of research (*Will et al., 2015*; *Winardhi et al., 2015*). Because the anti-silencing style of gene regulation is indirect and does not rely on specific DNA interactions, searching for SsrB binding sites has not been informative in uncovering this type of regulation (*Tomljenovic-Berube et al., 2010*; *Worley et al., 2000*; *Shea et al., 1996*). Even a recent report in which the proteomes of wild type, *hilA* null (a transcriptional regulator of SPI-1 genes) and *ssrB* null were analyzed by SILAC and compared with an existing CHIP dataset failed to identify *csgD* as an SsrB-regulated locus (*Brown et al., 2014*), as sequence gazing alone does not help in identifying mechanisms of transcriptional regulation.

SsrB is well suited to this style of regulation, because it does not recognize a well-defined binding site (*Feng et al., 2004*; *Walthers et al., 2007*; *Tomljenovic-Berube et al., 2010*), it has a high non-specific binding component (*Carroll et al., 2009*) and it bends DNA upon binding (*Carroll et al., 2009*; *Figure 6B*, this work). Furthermore, previous microarray studies disrupted both *ssrA* and *ssrB*, which would not uncover a distinct role for SsrB in gene regulation under non SPI-2-inducing conditions in the absence of the SsrA kinase. It is worth mentioning here that in our AFM images, it was apparent that H-NS was still bound to some regions of the *csgD* promoter when SsrB condensed the DNA (*Figure 6A(ii)*). Thus, H-NS does not have to be completely stripped off the DNA for de-repression to occur, a finding that was also evident in our previous studies (*Liu et al., 2010*) and others (*Will et al., 2014*).

SsrB binds and bends DNA, resulting in highly curved DNA conformations. This DNA binding property of SsrB is distinct from H-NS, which forms rigid nucleoprotein filaments and thus straight DNA conformations (*Figure 6A(i)*). Bent DNA is therefore an energetically unfavorable substrate for H-NS binding, and a likely mechanism of SsrB-mediated anti-silencing of H-NS repressed genes. SsrB-dependent displacement of H-NS is more energetically favored to occur predominantly at the ends of H-NS-bound filaments, which requires disruption of fewer H-NS protein-protein interactions (*Winardhi et al., 2015* and *Figure 6B*). In an equal mixture of H-NS and SsrB (*Figure 6A(ii)*), we do not see evidence of sharply bent filaments. This is expected because H-NS dissociation is likely restricted to the filament ends. Such events occur due to the cooperative nature of H-NS binding that results in a chain of linked H-NS proteins. Hence, H-NS displacement by SsrB likely occurs progressively from the filament end. This behavior has been observed in our single-molecule stretching experiments with H-NS filaments in the presence of SsrB. This ability of H-NS to re-orient on the DNA without being released would also promote its re-binding and silencing when SsrB or other anti-silencers are released (*Figure 6B*).

## Structural homology does not indicate functional homology

Response regulators are grouped into subfamilies on the basis of the structures of their DNA binding domains. SsrB is in the NarL/FixJ subfamily, which possess a helix-turn-helix (HTH) motif in the C-terminus (*Baikalov et al., 1996*). NarL was the first full-length structure of a response regulator and it showed that the N-terminal phosphorylation domain physically blocked the recognition helix in the HTH motif (*Maris et al., 2002*). Thus, phosphorylation is required to relieve the inhibition of the N-terminus. In the results presented herein, it is apparent that SsrB has adapted to relieving H-NS-silencing and that phosphorylation is not required for this behavior, nor is it required for DNA binding (*Figure 5B*).

In summary, we showed that the response regulator SsrB is required for biofilm formation because it can de-repress H-NS at the *csgD* promoter (*Figure 6B*). This leads to the production of CsgD, the master regulator of biofilms. It is noteworthy that a laterally acquired gene product, SsrB, has evolved the job of regulating the levels of *csgD,* a transcriptional regulator encoded by the core genome. For this activity, phosphorylation of SsrB was not required, which is rare amongst response regulators. Furthermore, we identify H-NS as a repressor of *csgD* in *Salmonella*, instead of an activator (*Gerstel et al., 2003*). This unifies the regulation of CsgD by H-NS in *E. coli* (*Ogasawara et al.,*

**Table 1.** List of Bacterial strains and plasmids.

| Strain | Description/Nomenclature | Reference |
|---|---|---|
| wild type | *Salmonella enterica* serovar Typhimurium strain 14028s | Lab strain collection |
| ssrA | *ssrA::TetRA* derivative of 14028s | This work |
| ssrB | DW85 | Don Walthers (originally from Stephen Libby) |
| ssaC | *ssaC::TetRA* derivative of 14028s | *Chakraborty et al. (2015)* |
| ssaJ | *ssaJ::TetRA* derivative of 14028s | Hideaki Mizusaki unpublished |
| D56A | D56A SsrB derivative of 14028s | This work |
| ssrAH1 | DW748 | Don Walthers unpublished |
| ssrAD | DW749 | Don Walthers unpublished |
| ssrAH2 | DW750 | This work |
| ssrAH1 sifA-LacZ | Made by transducing *sifA-lacZ* from DW636 to DW748 | This work |
| ssrAD sifA-lacZ | Made by transducing *sifA-lacZ* from DW636 to DW749 | This work |
| ssrAH2 sifA-lacZ | Made by transducing *sifA-lacZ* from DW636 to DW750 | Don Walthers; Lab strain collection |
| DW636 | *sifA-lacZ* at *attP* site in 14028s | Don Walthers; Lab strain collection |
| DW637 | *ssrB::Km* derivative of DW636 | This work/Don Walthers (lab strain collection) |
| ssrAH1 AcP | *ackA-pta::Km* (from DW142) transduced in DW748 | This work/Don Walthers (lab strain collection) |
| ssrAD AcP | *ackA-pta::Km* (from DW142) transduced in DW749 | This work/Don Walthers (lab strain collection) |
| ssrAH2 AcP | *ackA-pta::Km* (from DW142) transduced in DW750 | This work/Don Walthers (lab strain collection) |
| hns | *hns::TetRA* derivative of DW636 | This work/Don Walthers (lab strain collection) |
| hns ssrB | *hns::TetRA* derivative of DW637 | This work/Don Walthers (lab strain collection) |
| Plasmid pKF46 | D56A His-SsrB pMpM-A5Ω construct | *Feng et al. (2004)* |
| Plasmid pKF43 | His-SsrB pMpM-A5Ω construct | *Feng et al. (2004)* |
| Transformant DW160 | DH5α harboring His-HNS (*S. typhimurium*) in pMpM-A5Ω | *Walthers et al. (2011)* |
| Plasmid pKF104 | His-SsrBc pMpM-A5Ω construct | *Feng et al. (2004)* |
| pBR328::csgD | *csgD* construct | Prof Iñigo Lasa's group |
| Plasmid pRC24 | K179A His-SsrB pMpM-A5Ω construct | *Carroll et al. (2009)* |

*2010*) and *Salmonella*. This work places SsrB at a unique decision point in the choice between life-styles by *Salmonella* and makes it crucial for the entire gamut of pathogenesis, i.e., biofilms and virulence.

## Materials and methods

### Bacterial growth and media
The bacterial strains and plasmids used in this study are listed in *Table 1*. *Salmonella enterica* serovar Typhimurium strains were grown in LB medium with shaking at 37°C in the presence of 100 μg/ml ampicillin, 12.5 μg/ml Tetracycline (Tet) or 50 μg/ml Kanamycin (Km) when necessary. For observing the rdar morphotype, plates of LB medium (without salt) containing 1% Tryptone and 0.5% Yeast Extract supplemented with congo red (40 μg/ml) (Sigma-Aldrich, Singapore) were prepared and kept at 30°C after inoculation. To detect cellulose production, LB without salt medium plates were supplemented with Fluorescent Brightener 28 (200 μg/ml) (Sigma-Aldrich), stored under darkness and observed under UV light after incubating at 30°C. For the SsrBc complementation test using the plasmid pKF104, 0.2% arabinose was added to the medium.

### Molecular biology techniques
All DNA manipulation procedures were carried out according to (*Sambrook, 1989*) using reagents procured from Qiagen, Invitrogen or Fermentas, Singapore. Transformation in *S.* Typhimurium

**Table 2.** List of oligonucleotides.

| Purpose/name | Sequence (5'-3') |
| --- | --- |
| Digf (forward 755bp *csgD* regulatory region) | tgatgaaactccactttttta |
| Digr (reverse 755bp *csgD* regulatory region) | tgctgtcaccctggacctggtc |
| *ssrA* knockout (forward) | atgaatttgctcaatctcaagaatacgctgc aaacatctt ttaagacccactttcacatt |
| *ssrA* knockout (reverse) | agccgatacggcattttcaatatcagccag caagaggtcc ctaagcacttgtctcctg |
| csg1 (forward *csgD* internal) | ggaagatatctcggccggttgc |
| csg2 (reverse *csgD* internal) | tcagcctagggataatcgtcag |
| rrsA1 (forward *rrsA* internal) | gcaccggctaactccgtgcc |
| rrsA2 (reverse *rrsA* internal) | gcagttccaggttgagcccg |
| PSsrBF (forward for pKF46) | atgaaagaatataagatcttat |
| PSsrBTR (hybrid reverse for pKF46) | ttaatactctaattaacctcattcttcgggcac agttaagtctaagcacttgtctcctg |
| TSsrBF (forward TetRA-*ssrB*) | acttaactgtgcccgaagaatgaggttaata gagtattaattaagacccactttcacatt |
| TSsrBR (reverse TetRA- after *ssrB* stop) | catcaaaatatgaccaatgcttaataccatc ggacgcccctggctaagcacttgtctcctg |
| Digb (Forward for EMSA) | Biotin- tgatgaaactccactttttta |
| CsgDigRS (Reverse for EMSA) | aatattttctctttctggata |
| *hns* knockout (forward) | gctcaacaaaccaccccaatataagtttgg attactacattaagacccactttcacatt |
| *hns* knockout (reverse) | atcccgccagcggcgggattttaagcatcc aggaagtaaactaagcacttgtctcctg |

strains was performed by standard electroporation protocols (*Sambrook, 1989*). Polymerase chain reaction (PCR) was carried out using oligonucleotides as listed in the *Table 2* following standard protocols (*Sambrook, 1989*).

## Strain construction

The *S.* Typhimurium strain harboring a D56A mutation in SsrB was generated by the homologous recombination technique as decribed in (*Karlinsey, 2007*). Plasmid pKF46 (*Feng et al., 2004*) was used to construct the necessary linear DNA fragment by a two-step overlapping PCR (*Sambrook, 1989*). Homologous recombination was also used to construct the *hns* and *ssrA* deletion mutants in the respective strain backgrounds. The *sifA-lacZ* marker was transduced into the lambda *attP* site of strains DW748 (ssrAH1), DW749 (ssrAD) and DW750 (ssrAH2) using standard P22 transduction protocols (*Davis, 1980*).

## Fluorescence confocal imaging of flow cell biofilms

Biofilms were grown in three-channel glass-bottomed flow cells (individual cell dimensions $1 \times 4 \times 40\ mm^3$) with an individual-channel flow rate of 8 ml/h of 0.5X M9 glucose minimal medium. Each channel was inoculated with M9 minimal medium cultures adjusted to an $OD_{600}$ of 0.04. Flow was started after allowing initial attachment for 1 hr. After specific time points, each channel of the flow cell was stained with 200 µl SYTO-9 green (Invitrogen, Singapore) as per the manufacturer's protocol. Subsequently, for each flow cell channel image acquisition was performed using a LSM 780 Carl Zeiss confocal fluorescence microscope at 480/500 nm (20X magnification). Five image stacks were acquired starting from the center of the channel to a distance of 5–10 mm from the inlet, approximately 5 mm apart. This experiment was repeated twice in duplicates for the wild type and *ssrB* mutant strains. All images were processed using the Image J software.

## Scanning electron microscopy (SEM)

Static biofilms were grown by inoculating the bacterial strains on APTES-coated coverslips in 24-well Nunclon polystyrene plates in 500 µl LB without salt medium and kept shaking at 30°C, 100 rpm. After two days, the growth medium was removed; wells were washed with PBS and incubated with 200 µl of the fixative solution (4% para-formaldehyde+0.2% glutaraldehyde in filter-sterilized PBS) for 1 hr at room temperature. After washing twice with sterile water, 35%, 50%, 75%, 90%, 95% and 100% ethanol were added sequentially and kept for 10 min at room temperature. Dehydrated samples were stored in absolute ethanol and kept at 4°C until subjected to critical point drying. Images were obtained using a Carl Zeiss Merlin field emission scanning electron microscope by detecting secondary electrons (SE2) under low electron beam acceleration voltage (1 kV) and low probe current (76 pA).

## Crystal violet staining

To estimate the amount of biofilms typically a single colony was inoculated in LB broth medium and incubated at 37°C/250 rpm. 2 µl of this culture was added to 198 µl LB without salt medium in a 96-well polystyrene plate and kept at 30°C with gentle shaking of 100 rpm. After two days, the growth medium was removed and each well was washed twice with 200 µl of Phosphate-buffered Saline (PBS). The attached bacterial communities or biofilms in each well were then stained with 200 µl crystal-violet solution (0.1%) for a minimum of 1 min. This was followed by washing twice with PBS and addition of 200 µl absolute ethanol. Appropriate dilutions were measured for absorbance at 580 nm using a Tecan Infinite M200 plate reader. Each experiment was performed at least thrice in triplicates. For the time course experiment, the above procedure was performed twice at 12, 24, 36, 47, 60, 72 and 84 hr respectively in triplicates.

## Tube biofilm assay (TBA)

To study the capability of the *Salmonella* strains to form biofilms on cholesterol-attached surfaces, a tube biofilm assay was performed as described in (*Crawford et al., 2008*), but without the use of bile salts in the growth medium. LB without salt medium was used for growth at all steps. All experiments were performed at least thrice in triplicates for 7 days on a nutator shaker at room temperature. Biofilms formed at the end of seven days were estimated using the above described crystal violet staining protocol with appropriate controls.

## Beta-galactosidase assay

Each of the bacterial strains were grown overnight in LB, washed twice and resuspended in 50 µl of PBS buffer. These were then transferred into 1 ml of fresh MgM medium, pH 5.7, as in (*Feng et al., 2003*) and kept shaking at 37°C for around 6–7 h, until growth reached $OD_{600}$ between 0.6–0.8. At this stage, 50 µl of the culture was removed in a 96-well microtiter plate and 145 µl of lysis buffer was added (0.01% SDS, 50 mM Beta Mercaptoethanol in Z-buffer) as performed previously (*Feng et al., 2003*). The β-galactosidase activity was represented in Miller Units and calculated as $1000 \times [(OD_{420}\text{-}1.75 \times OD_{550})]/ t (min) \times volume (ml) \times OD_{600}$. Measurements were made in a Tecan Infinite M200 plate reader and repeated thrice in triplicates.

## RNA isolation

Total RNA was isolated from two-day old macrocolonies using the Qiagen RNeasy kit (Qiagen, Singapore) as per the manufacturer's protocol with some modifications. Briefly, the macrocolonies were selected and re-suspended in 1 ml Qiagen RNAprotect reagent followed by immediate total RNA isolation or storage at -80°C. Appropriate volume of lysis buffer, 15 mg/ml lysozyme in TE buffer, was used. After dissolving the RNA in 20 µl RNase-free water, samples were assessed by gel electrophoresis and quantified using the Nanodrop system (THERMO Scientific, Singapore). All RNA samples were treated with TURBO DNase as per the manufacturer's protocol (Life Technologies, Singapore). PCR analysis using *rrsA* specific primers was performed to ensure the absence of genomic DNA in the RNA preparations.

## Reverse transcription and quantitative real-time PCR

Reverse transcription reaction was carried out using the iScript reverse transcription supermix for RT-qPCR (BIO-RAD, Singapore) according to the recommended protocol. This was followed by amplifying 5–10 ng cDNA by real-time qPCR using SsoFast EvaGreen Supermix (BIO-RAD) and *csgD*-specific primers. Similar reactions were set up using *rrsA*-specific primers for their use as normalization controls. All experiments were performed in triplicate with two independent RNA preparations. Relative transcript abundance was determined using the $\triangle C_T$ method using a reference gene as per the manufacturer's guide.

## Immunoblot analysis

Two-day old macrocolonies were resuspended in 400 µl Laemmli buffer (*Sambrook, 1989*) and total protein was separated by 10% SDS-PAGE. This was followed by electro-transfer to a PVDF membrane as described before (*Feng et al., 2004*). The membrane was incubated with anti-CsgD (1:10,000) or anti-GroEL (1:5000) antibodies in PBS buffer containing 0.05% Tween-20 and 3% BSA. Anti-rabbit secondary antibody (Santa Cruz Biotechnology Inc., Dallas, TX) was used for detection as described previously (*Feng et al., 2004*).

## Overexpression and purification of proteins

The *E. coli* BL21 (DE3) strain was used as a host for the overproduction of proteins His-SsrBc, His-SsrB, D56A His-SsrB, K179A His-SsrB and His-H-NS. The respective plasmids harboring the constructs have been listed in Supplementary *Table 2*. Detailed procedures for the purification of His-SsrBc, His-SsrB, D56A His-SsrB and K179A His-SsrB have been described before (*Feng et al., 2004*; *Walthers et al., 2007*; *Carroll et al., 2009*). The DW160 plasmid harboring His-tagged H-NS from *Salmonella* Typhimurium was used to overexpress and purify His-H-NS following the procedure described in (*Walthers et al., 2011*).

## Atomic force microscopy (AFM)

Glutaraldehyde-modified mica surface was prepared as decribed previously in (*Liu et al., 2010*) and references therein. A 755 bp sequence upstream to the +1 start site (*Gerstel et al., 2003*) of *csgD* was amplified and gel purified. This fragment also harbored a SsrB-specific site (*Feng et al., 2004*; *Walthers et al., 2007*), TTATAAT sequence (*Figure 6—figure supplement 4*). A typical 50 µl reaction contained 10 ng of this DNA (755 bp of the *csgD* regulatory region) mixed with an appropriate amount of SsrB, SsrBc or D56A SsrB and incubated for 15 min at room temperature. This mixture was then deposited on glutaraldehyde-modified mica for 15 min. Images were acquired on a Bruker Dimension FastScan AFM system using the tapping mode with a silicon nitride cantilever (FastScan C, Bruker). Raw AFM images were processed using Gwyddion software (http://gwyddion.net/). The bending angle was analysed using a home-written Matlab code as described in the Supplementary methods.

## Statistical analysis

GraphPad Prism 6 version 6.00, GraphPad Software, La Jolla California, www.graphpad.com was used to make all the graphs and do statistical analysis by Student's t-test wherever required.

## Supplementary methods

### Growth curves (Total viable count)

*Salmonella enterica* serovar Typhimurium strains were grown in LB medium overnight from single colonies at 37°C/250 rpm. 1% of starting cultures were added to LB (without salt) medium containing 1% Tryptone and 0.5% Yeast Extract and incubated shaking at 37°C/100 rpm in 14 ml tubes. At every 2 hr, an aliquot from the culture was removed, diluted appropriately and plated on LB agar plates. The number of colonies formed after incubation at 37°C were counted and growth curves were plotted.

### Estimation of adherent and non-adherent sub-populations

*Salmonella enterica* serovar Typhimurium strains were grown in 50 ml 1% Tryptone medium in 250 ml flasks and kept shaking at 28°C/200 rpm (*MacKenzie et al., 2015*). After 2 days, the entire 50 ml

culture was transferred in 50 ml falcon tubes, including the aggregates that have attached to the wall of the flask, and was centrifuged at 210 rpm/2 min. Appropriate dilutions from the supernatant or the non-adherent (free-swimming) fraction were plated on LB agar plates for enumeration. Total wet weight of the pellet or the adherent (multicellular) fraction was determined after a brief centrifugation at 10,000 rpm for 1 min.

## Electrophoretic mobility super shift assay

Electrophoretic mobility assay using the LightShift Chemiluminescent EMSA kit (THERMO Scientific) was modified to detect complexes formed by the binding of SsrB and/or H-NS to the *csgD* regulatory region. A 5'-biotinylated *csgD* DNA fragment was prepared by amplifying the upstream regulatory region of *csgD* (122bp) using respective primers from the wild type strain 14028s. 5 fmol of this probe was then incubated with 1.08 μM SsrB/K179A SsrB or 15 μM H-NS in a binding buffer containing 10 mM Tris, pH 7.5; 50 mM KCl, 2.5% glycerol; 0.05% Nonidet P-40 and 1 μg poly (dI-dC), for 20 min at RT. In case where competitor DNA was added, a 200-fold excess of an unlablelled $csgD_{122}$ was first incubated with 1.08 μM SsrB for 20 min, followed by the addition of 5 fmol labelled probe to the reaction buffer and subsequent incubation for 20 min. For detecting a supershift, 1.08 μM SsrB and 15 μM H-NS were incubated with 0.5 μl anti-SsrBc serum in the binding buffer for 1 hr at RT. 5 fmol of the probe was then added and incubated for additional 20 min at RT. The reactions were subsequently subjected to electrophoresis, electro-blotting, cross-linking and detection as described previously (*Carroll et al., 2009*).

## Measurement of the bending angle (*Figure 7*).

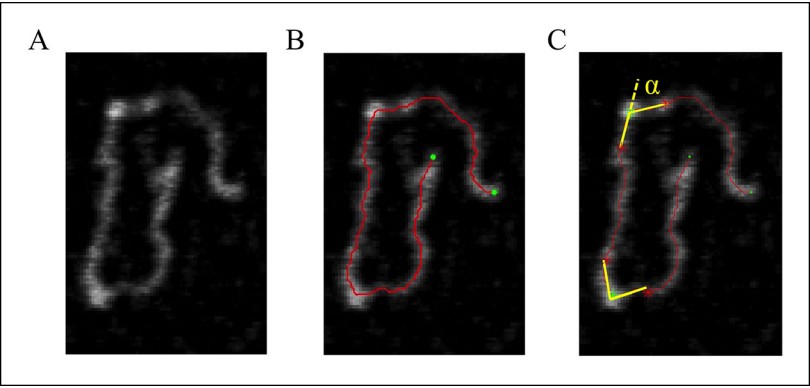

**Figure 7.** DNA tracing and bending angle measurement. (A) The original AFM image as processed with the Gwyddion software. (B) A Matlab code was used to trace the DNA in the AFM images. The digitized binary line represents the DNA backbone (red line), with the two end points marked by green dots. (C) For the bending angle measurement, a point along the red line was manually selected as the binding location of SsrB on DNA (green asterisk). The red asterisks indicate two points that are 15 nm upstream and downstream from this protein-binding site. Linear interpolation of points scattered along the green and red asterisk was used to plot a straight line between these points (yellow line). The angle between the yellow lines (α) is determined as the bending angle of SsrB; 81 such measurements were made. The same procedure was followed for the control AFM images of $csgD_{755}$, 327 such measurements were made.

## Liquid atomic force microscopy (AFM)

Glutaraldehyde-modified mica surface was prepared as decribed previously in (*Liu et al., 2010*) and references therein. A 755 bp sequence upstream to the +1 start site (*Gerstel et al., 2003*) of *csgD* was amplified and gel purified. This fragment also harbored a SsrB-specific site (*Feng et al., 2004*; *Walthers et al., 2007*), TTATAAT sequence (*Figure 6—figure supplement 4*). A typical 50 μl reaction contained 10 ng of this DNA (755 bp of the *csgD* regulatory region) mixed with an appropriate amount of SsrB or H-NS and incubated for 15 min at room temperature. This mixture was then deposited on glutaraldehyde-modified mica for 15 min. Images were acquired on a Bruker Dimension FastScan AFM system using the tapping mode with a silicon nitride cantilever (FastScan C, Bruker). Raw AFM images were processed using Gwyddion software.

## Acknowledgements

We thank the following: Assoc Prof Scott Rice, SCELSE, Singapore, for his guidance with the flow cell experiments, Prof T Venky Venkatesan, NUSNNI, Singapore, for the use of SEM facilities, Dr. Aaron White, Vaccine and Infectious Disease Organization, University of Saskatchewan, Canada, for the monoclonal anti-CsgD antibody, Prof Iñigo Lasa, Universidad Pública de Navarra-CSIC-Gobierno de Navarra, Spain, for the *csgD* construct and Dr. Swaine Chen, GIS, Singapore, for the anti-GroEL antibody. We are grateful to the MBI Central facilities, especially Chen HongYing from the Protein Expression core and Zhang Bo, Cindy, from the Science Communications core for the artwork. We appreciate Prof Stephen Lory, Harvard Medical School, Boston, and the three anonymous reviewers for their critical comments. Supported by 5IOBX-000372 from the VA to LJK, Singapore Ministry of Education Academic Research Fund Tier 2 [MOE 2013-T2-1-154] to YJ and LJK and funding from the Research Centre of Excellence in Mechanobiology, NUS from the Ministry of Education.

## Additional information

### Funding

| Funder | Grant reference number | Author |
| --- | --- | --- |
| VA | 5IOBX-000372 | Linda J Kenney |
| Research Centre of Excellence in Mechanobiology, NUS, Ministry of Education, Singapore | | Linda J Kenney |
| Singapore Ministry of Education Research Fund, Tier 2 | MOE 2013-T2-1-154 | Yan Jie<br>Linda J Kenney |

The funders had no role in study design, data collection and interpretation, or the decision to submit the work for publication.

### Author contributions

SKD, Conception and design, Acquisition of data, Analysis and interpretation of data, Drafting or revising the article; RSW, Acquisition of data, Analysis and interpretation of data, Drafting or revising the article; SP, Conception and design, Acquisition of data; MMD, Acquisition of data, Drafting or revising the article; YJ, Analysis and interpretation of data, Drafting or revising the article; LJK, Conception and design, Analysis and interpretation of data, Drafting or revising the article

### Author ORCIDs

Michal M Dykas, http://orcid.org/0000-0001-5933-314X

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
