## [Decision Letter]

Thank you for submitting your work entitled "To P or not to P a role for unphosphorylated SsrB in activating *csgD (agfD*) to regulate *Salmonella* Typhimurium biofilms" for consideration by *eLife*. Your article has been reviewed by three peer reviewers, one of whom is a member of our Board of Reviewing Editors, and the evaluation has been overseen by Richard Losick as the Senior Editor.

The reviewers have discussed the reviews with one another and the Reviewing Editor has drafted this decision to help you prepare a revised submission.

This is an interesting paper that describes an unexpected role for a *Salmonella* response regulator in a two-component regulatory system. The authors conclude that the response regulator SsrB has a gene regulatory function in addition to its established role in accepting phosphate from SsrA to activate so-called SPI-2 genes important for virulence. In an SsrA-independent manner SsrB derepresses transcription of *csgD*, which itself serves as a transcription activator of genes required for biofilm formation. In this way SsrB can serve as a decision point between the planktonic and biofilm lifestyles this bacterium has in colonized hosts. The manuscript shows that *csgD* activation involves unphosphorylated SsrB. Evidence is consistent with the idea that HNS binds the *csgD* promoter region and this silences or represses *csgD* expression. Unphosphorylated SsrB derepresses *csgD* not by binding HNS but by binding to the *csgD* promoter itself and occluding some of the HNS binding.

The reviewers believe that several modifications to the manuscript are needed prior to acceptance:

The title needs to be changed. The manuscript needs to be condensed. Some specific suggestions are provided in the detailed reviewer comments. There is thought that Figure 1 is not relevant. Reviewer 2 has some specific concerns about Figure 2, Figure 4 and Figure 6 that require some attention and Figure 6—figure supplement 1 requires additional controls. There is some concern about whether the artificial in vitro systems reflect the native situation. Some caution about interpretation is advised.

Reviewer 3 would like to see additional biofilm data, "It is therefore necessary to somehow quantify the entire population of Salmonellae in the tube and assess what fraction of it adheres. A true biofilm-defective mutant should show a shift from a fraction that adhere to the non-adhering planktonic population." Finally, Reviewer 3 points out that, "The elegant experiments using AFM need to be supported with certain controls, primarily aimed at demonstrating specificity of binding of SsrB to the *csgD* regulatory region (and lack of binding to other H-NS silenced regions) in both AFM and electrophoretic mobility shift assays.

*Reviewer #1:*

This is an interesting paper that describes an unexpected role for a *Salmonella* response regulator in a two-component regulatory system. The paper shows that the response regulator SsrB has a gene regulatory function in addition to its established role in accepting phosphate from SsrA to activate so-called SPI-2 genes important for virulence. In an SsrA-independent manner SsrB derepresses transcription of *csgD*, which itself serves as a transcription activator of genes required for biofilm formation. In this way SsrB serves as the decision point between the planktonic and biofilm lifestyles this bacterium has in colonized hosts. The manuscript shows that *csgD* activation involves unphosphorylated SsrB. It shows that HNS binds the *csgD* promoter region and this silences or represses *csgD* expression. Unphosphorylated SsrB derepresses *csgD* not by binding HNS but by binding to the *csgD* promoter itself and occluding some of the HNS binding. This was shown by AFM of single molecule binding to DNA. The microscopy is well suited for *eLife* publication.

The work is interesting in a number of aspects. It points to novel ways in which response regulator components of two-component regulatory systems can function. It explains for the case of one of the still rare examples where an unphosphorylated response regulator can function how it does so. It addresses the issue of the important lifestyle "decision" that *Salmonella* makes. It also provides insight about HNS silencing and how anti-silencing can occur. This is a current area of interest with respect to evolution of bacterial pathogens.

There are several issues with the manuscript that the authors need to address. I present them here in the order in which they arise in the manuscript.

1) Title. "To P or not to P" conveys little information and might be appropriate for the Insight published along with the article. The rest of the title will make sense only to those working in the field. I would suggest something along the lines of "An unphosphorylated two-component response regulator induces *Salmonella* Typhimurium biofilm genes by antisilencing a biofilm master-regulator gene."

2) Key words. I think key words are meant to exclude words already appearing in a title.

3) Abstract. Delete "Remarkably." There is an overuse of emphasis words in general in the manuscript. I find this quite distracting. "Highly upregulated", unequivocally rule out, "Sharp contrast", etc. These all seem like overstatements to me.

4) The Introduction is very long. It reads like the introduction to a thesis. I am not sure what might be condensed but it would be helpful to condense if possible.

5) Introduction, fourth paragraph. Delete "Walthers and Kenny unpublished." It is not something the reader can refer to as are the publications referenced.

6) Last two paragraphs of the Introduction. As written it is confusing as to what the authors showed previously and what they are showing in this manuscript.

7) Subsection “Unphosphorylated SsrB activates biofilm formation”. I think the interpretations should be recast. First we learn that SsrA is not required for the biofilm phenotype. So it could be unphosphorylated B or B could be phosphorylated by an alternate donor. Then we learn that general donors like acetyl-P are not involved. So it could be unphosphorylated or there could be some sort of unknown cross talk with another sensor kinase. Finally we learn that Asp56, the residue phosphorylated by SsrA, can be mutated to an alanine and the protein still functions with respect to biofilm formation (but not SPI2 gene activation). This is good genetic evidence that the protein induces biofilm formation in an unphosphorylated form.

8) Subsection “Unphosphorylated SsrB activates biofilm formation”. This is an example of a sentence that needs rewriting. "To confirm our hypothesis we examined…". This is not the scientific method. Experiments are meant to be designed to disprove one’s hypothesis. A simple wording change would provide a fix, for example: "To test our hypothesis…". This wording problem comes up in a few other places as well.

9) Subsection “Unphosphorylated SsrB activates biofilm formation”. Change "This is the first evidence" to "This provides evidence that".

10) Subsection “SsrB and H-NS differentially regulate *csgD* expression”, third paragraph. I am missing something here. I don't understand why the hypothesis explains why phosphorylation was not required.

11) Subsection “SsrB binds an H-NS stiffened nucleoprotein filament at *csgD*”. The last sentences seem to describe a model based on the evidence presented rather than a proven mechanism and should be presented in this way. Particularly for the last sentence on RNAP I see no evidence in the manuscript.

12) Subsection “SsrB sits at a pivotal decision point that determines *Salmonella* lifestyles”, first paragraph and elsewhere. "It can de-repress HN-S at *csgD*" seems like confusing jargon to me. Do the authors mean "it can counter HN-S repression of *csgD*" or "counter HN-S silencing of”?

13) Subsection “SsrB sits at a pivotal decision point that determines *Salmonella* lifestyles”. Again I did not see any experiments about enabling access of RNAP. This seems logical but it should be presented as a model.

14) The model is presented as a counterpoint to the idea that periplasmic histidines can play a role ("we think this is unlikely"). I got a little confused here about whether maybe both things could be true. Are they mutually exclusive?

15)”It is worth mentioning here that in our AFM images, it was apparent that H-NS was still bound to some regions of the *csgD* promoter when SsrB is condensing the DNA (Figure 6(ii))”. There is a mixture of tenses. I would change "when SsrB is condensing" to "was condensing."

16) It seems like the whole first paragraph of the subsection “Structural homology does not indicate functional homology “could be deleted. I am not engaged by the fact that "we were lulled into thinking that SsrB would behave like NarL." There are examples of transcriptional regulator families where some members require ligand binding to bind DNA and activate genes. They do this in the same way that NarL interacts with P to expose the DNA binding motif. Other members of the same family bind DNA and repress genes in the absence of ligand. The ligand blocks DNA binding.

The last half of the last paragraph of the Discussion is maybe the fourth or fifth time the authors bring up the fact that their conclusion that HNS represses or silences *csgD* conflicts with the conclusion of Gerstal. Although this may be true, the way it is presented is not constructive. First, I believe this point should not be raised in the Results (as it is now). It is a point of discussion. Second, I don't think, as it stands, it corrects the scientific record. If the authors want to discuss this they should make some attempt to resolve the conflict. Why do they think they arrived at a conclusion opposing that of Gerstel et al.? Are there strain differences? Did Gesrtel misinterpret or over-interpret a result? If the authors want to put any focus on the different conclusions then they should try to help us understand what might be a possibility. It is ok to say something about how at least with our strains and experimental protocols we find HNS silencing of *csgD* whereas Gerstel using a different whatever found HNS activation or whatever it is. It takes a level of brio to just claim ones results are more believable than others.

*Reviewer #2:*

In this manuscript the authors build on previous work demonstrating that the SsrAB regulators are important to *Salmonella* typhimurium biofilm formation. They determine that the SsrA histidine kinase is not required, but that the SsrB response regulator is required for biofilm formation as measured using methylene blue staining, Congo Red, and Calcofluor plates. They then analyzed the colonies formed in a microfluidic chamber in M9 medium in a flow cell and in cholesterol coated eppendorf tubes to further substantiate the biofilm phenotype. The authors provide support for the lack of phosphorylation of SsrB being what is necessary for SPI2 regulation of curli genes responsible for biofilms by using SsrB phosphorylation deficient bacteria, with mutations in SsrA and the ability to produce acetyl phosphate. The requirement of SsrB to form biofilms was repressed by deletion of H-NS providing in vivo evidence that SsrB could act by competing or silencing H-NS based repression. The authors then provide evidence that SsrB/HNS bind to the Csg promoter responsible for the synthesis of curli. The results add to our knowledge of *Salmonellae* gene regulation of biofilm formation, but do not really define where and how this may occur in vivo and/or what specific environmental conditions would lead to activation but not phosphorylation of SsrB.

Specific comments:

A great deal of data was not shown and this should be corrected prior to publication.

Throughout the manuscript the authors claim that they showed different results than that shown in Hamilton et al. However in this work the effect of an ssrA mutant on biofilm formation was complemented with a plasmid encoding both ssrA and ssrB. Therefore the results are likely a result of mutation of both genes or polarity. Therefore the results are really similar and the authors have provided molecular detail as to the mechanism.

Since most of the paper is about mechanisms of transcription that lead to biofilm formation I am not sure what Figure 1 adds to the manuscript.

In Figure 2 the staining seems the same at 6 days. This is a surprising result, has the biofilm been formed that rapidly by another sensing mechanism?

In Figure 2 why does the increase at later time points occur and why is it greater at 36 hours than at 47 and 69 hours? How many replicated experiments are represented by the figure and are the statistical analyses significant? This is not commented on in the figure legend.

The phenotype of the *ssrA* mutant should be shown in Figure 2.

The higher migrating signal in the western blot in Figure 4 is said to be non-specific in the figure legend. However, It is not present to a very great degree in the wild type and is greater in the *SseA* and D56A mutant and absent in *ssrB* mutants. What is this band and does it indicate another larger form of CsgD or has a different promoter been utilized when non-phosphorylated SsrB is present? This band should be isolated and protein sequenced to define its nature.

What is the evidence that the binding angle observed in vitro with an artificial substrate by ATM reflects its ability to promote anti-silencing (subsection “Unphosphorylated SsrB binds to the *csgD* regulatory region”, first paragraph)?

Again condensation on a silica binding substrate of a regulatory region on an increase of concentration of the protein that binds may or may not be biologically relevant in vivo (subsection “Unphosphorylated SsrB binds to the *csgD* regulatory region”, last paragraph). The authors should reduce this speculation and consider performing the AFM in solution and under helium rather than nitrogen and see if similar results are obtained.

The fact that HNS alters structure of DNA on a solid support does not reaffirm more than the biological data the mechanism by which HNS represses transcription, it just indicates under these conditions in binds to the DNA (subsection “SsrB binds an H-NS stiffened nucleoprotein filament at *csgD*”, first paragraph).

I think the data support a model in which SsrB alters the bending of DNA, though it seems this may not be a direct effect in vivo and the authors should soften the statement that they have identified the mechanism by which unphosphorylated SsrB regulates *csgD*.

I am not sure the authors are correct in their ideas about SsrB function controlling extracellular lifestyle of *Salmonellae* since there are many growth conditions in which ssrAB are not expressed and the bacteria can form biofilms. Hence this may be most important for the expression of a biofilm under certain conditions of SPI2 expression but not SsrA activation as a kinase. This may actually occur inside host cells and recent evidence of cellulose expression in host cells and past work showing curli biosynthesis genes can be regulated by RpoS may suggest that they are expressed inside of host cells and hence the authors results may be most relevant to how curli may be expressed inside of host cells rather than SsrAB being another host switch more like PhoPQ.

The title contains this “to P or not to P”, which I found confusing and relatively unscientific since it does not represent PO4. This should probably be changed prior to publication.

In Figure 6—figure supplement 2, why is so little shifting seen with SsrB in the absence of HNS? Is the affinity very low? Additional controls should include SsrB without the anti-ssrB antibody that causes the super shift. Is no binding seen without HNS? If the AFM is significant, binding should be seen without HNS present.

The data would be strengthened by footprinting and defining the specific promotor sequences bound by SsrB and H-NS prior to publication.

Reviewer #3:

The authors present a series of experiments demonstrating a novel role for a two component regulatory system, implicated in the virulence and persistence of *Salmonella enterica*. The main novel aspects of the findings described in the manuscript are: 1) Demonstration that the SsrB response regulator is a molecular switch, reciprocally controlling two sets of traits associated with virulence (the expression of a type III secretion system) and biofilm formation; 2) Defining an active function for an un-phosphorylated response regulator; 3) Elucidation of a novel regulatory mechanism for a response regulator activity (anti-silencing) by DNA bending and 4) Evolutionary implications for acquisition of a regulatory system by horizontal gene transfer, controlling the expression of core, chromosomal genes.

Overall, this paper contains substantial amount of new results (in quality and in quantity) and represents information with broad interest to those readers who are interested in new molecular mechanisms of gene regulation, strategies of pathogenic organism capable of causing acute and chronic infections and their evolution. This is important work, established a new paradigm for two-component signaling. The only concern is with a few technical points related to certain experiments and a need for a few additional controls.

*Reviewer #3 (Minor comments):*

There are a few minor issue that the authors should address, mainly to make the manuscript more readable and highlighting its strongest points.

1) There is excessive amount of data on attempts to demonstrate that the absence of SsrB has an effect on biofilm formation. That could be simply reduced to a single figure comparing *ssrB* deletion with point mutants in SsrA and SsrB.

2) The major biofilm assay used in this work is based on adherence of bacteria to the sides of polystyrene tubes. This method actually measures only adhesion of the bacteria to their biofilm phenotype. A simple growth defect of the *ssrB* mutant would be also reflected as a decrease in adherence and erroneously concluded as a defect in biofilm formation. It is therefore necessary to somehow quantify the entire population of *Salmonellae* in the tube and assess what fraction of it adheres. A true biofilm-defective mutants should show a shift from a fraction that adhere to the non-adhering planktonic population.

3) Figure 4 is rather un-convincing; the lane containing the D56A mutant indicates some problems with transfer of CsgD from gels to the membrane and it does not correlate with the qRT-PCR data (Figure 4). The Western Blot should be either redone or left out.

4) The main issues with this work relates to the discovery of an anti-silencing mechanism by SsrB induced DNA bending. The elegant experiments using AFM need to be supported with certain controls, primarily aimed at demonstrating specificity of binding of SsrB to the csgD regulatory region (and lack of binding to other H-NS silenced regions) in both AFM and electrophoretic mobility shift assays.

---

## [Author Response]

*[…] The title needs to be changed. The manuscript needs to be condensed. Some specific suggestions are provided in the detailed reviewer comments. There is thought that Figure 1 is not relevant. Reviewer 2 has some specific concerns about Figure 2, Figure 4 and Figure 6 that require some attention and Figure 6—figure supplement 1 requires additional controls. There is some concern about whether the artificial invitro systems reflect the native situation. Some caution about interpretation is advised.*

The title has been changed and the manuscript has been condensed. Figure 1 has been removed and we have supplied the additional controls as requested.

*Reviewer 3 would like to see additional biofilm data, "It is therefore necessary to somehow quantify the entire population of Salmonellae in the tube and assess what fraction of it adheres. A true biofilm-defective mutant should show a shift from a fraction that adhere to the non-adhering planktonic population." Finally, Reviewer 3 points out that, "The elegant experiments using AFM need to be supported with certain controls, primarily aimed at demonstrating specificity of binding of SsrB to the csgD regulatory region (and lack of binding to other H-NS silenced regions) in both AFM and electrophoretic mobility shift assays.*

These experiments are now provided.

Reviewer #1:

*[…] There are several issues with the manuscript that the authors need to address. I present them here in the order in which they arise in the manuscript. 1) Title. "To P or not to P" conveys little information and might be appropriate for the Insight published along with the article. The rest of the title will make sense only to those working in the field. I would suggest something along the lines of "An unphosphorylated two-component response regulator induces Salmonella Typhimurium biofilm genes by antisilencing a biofilm master-regulator gene."*

The title has been changed to: “The horizontally-acquired response regulator SsrB drives a *Salmonella* lifestyle switch by relieving silencing a biofilm master regulator”.

*2) Key words. I think key words are meant to exclude words already appearing in a title.*

Redundant key words have been removed, although we left in two-component regulatory system.

*3) Abstract. Delete "Remarkably." There is an overuse of emphasis words in general in the manuscript. I find this quite distracting. "Highly upregulated", unequivocally rule out, "Sharp contrast", etc. These all seem like overstatements to me.*

We have toned down the use of emphasis words.

*4) The Introduction is very long. It reads like the introduction to a thesis. I am not sure what might be condensed but it would be helpful to condense if possible.*

The Introduction has been condensed.

5) Introduction, fourth paragraph. Delete "Walthers and Kenny unpublished." It is not something the reader can refer to as are the publications referenced.

We have removed this reference.

*6) Last two paragraphs of the Introduction. As written it is confusing as to what the authors showed previously and what they are showing in this manuscript.*

This section has been reworded.

*7) Subsection “Unphosphorylated SsrB activates biofilm formation”. I think the interpretations should be recast. First we learn that SsrA is not required for the biofilm phenotype. So it could be unphosphorylated B or B could be phosphorylated by an alternate donor. Then we learn that general donors like acetyl-P are not involved. So it could be unphosphorylated or there could be some sort of unknown cross talk with another sensor kinase. Finally we learn that Asp56, the residue phosphorylated by SsrA, can be mutated to an alanine and the protein still functions with respect to biofilm formation (but not SPI2 gene activation). This is good genetic evidence that the protein induces biofilm formation in an unphosphorylated form.*

We have presented these experiments with the same logic in the given order.

8) Subsection “Unphosphorylated SsrB activates biofilm formation”. This is an example of a sentence that needs rewriting. "To confirm our hypothesis we examined…". This is not the scientific method. Experiments are meant to be designed to disprove one’s hypothesis. A simple wording change would provide a fix, for example: "To test our hypothesis…". This wording problem comes up in a few other places as well.

We have reworded these sentences.

*9) Subsection “Unphosphorylated SsrB activates biofilm formation”. Change "This is the first evidence" to "This provides evidence that".*

This sentence has been changed.

*10) Subsection “SsrB and H-NS differentially regulate csgD expression”, third paragraph. I am missing something here. I don't understand why the hypothesis explains why phosphorylation was not required.*

We rephrased this sentence to make it clear. Our earlier work with SsrB demonstrated that phosphorylation was required for in vitro transcription (Walthers et al., 2007; Walthers et al., 2011). This requirement is typical of response regulators that make direct contacts with RNAP (e.g. OmpR~P interacts with a-CTD; Russo et al., 1993).

*11) Subsection “SsrB binds an H-NS stiffened nucleoprotein filament at csgD”. The last sentences seem to describe a model based on the evidence presented rather than a proven mechanism and should be presented in this way. Particularly for the last sentence on RNAP I see no evidence in the manuscript.*

These sentences have been modified.

12) Subsection “SsrB sits at a pivotal decision point that determines Salmonella lifestyles”, first paragraph and elsewhere. "It can de-repress HN-S at csgD" seems like confusing jargon to me. Do the authors mean "it can counter HN-S repression of csgD" or "counter HN-S silencing of”?

We changed the wording.

*13) Subsection “SsrB sits at a pivotal decision point that determines Salmonella lifestyles”. Again I did not see any experiments about enabling access of RNAP. This seems logical but it should be presented as a model.*

We changed the wording here.

*14) The model is presented as a counterpoint to the idea that periplasmic histidines can play a role ("we think this is unlikely"). I got a little confused here about whether maybe both things could be true. Are they mutually exclusive?*

This paragraph has been reworded.

*15)”It is worth mentioning here that in our AFM images, it was apparent that H-NS was still bound to some regions of the csgD promoter when SsrB is condensing the DNA (Figure 6(ii))”. There is a mixture of tenses. I would change "when SsrB is condensing" to "was condensing."*

Done.

*16) It seems like the whole first paragraph of the subsection “Structural homology does not indicate functional homology “could be deleted. I am not engaged by the fact that "we were lulled into thinking that SsrB would behave like NarL." There are examples of transcriptional regulator families where some members require ligand binding to bind DNA and activate genes. They do this in the same way that NarL interacts with P to expose the DNA binding motif. Other members of the same family bind DNA and repress genes in the absence of ligand. The ligand blocks DNA binding.*

This paragraph has been modified.

*The last half of the last paragraph of the Discussion is maybe the fourth or fifth time the authors bring up the fact that their conclusion that HNS represses or silences csgD conflicts with the conclusion of Gerstal. Although this may be true, the way it is presented is not constructive. First, I believe this point should not be raised in the Results (as it is now). It is a point of discussion.*

We moved our description of this work to the Discussion.

*Second, I don't think, as it stands, it corrects the scientific record. If the authors want to discuss this they should make some attempt to resolve the conflict. Why do they think they arrived at a conclusion opposite that of Gerstel et al.? Are there strain differences? Did Gesrtel misinterpret or over-interpret a result? If the authors want to put any focus on the different conclusions then they should try to help us understand what might be a possibility. It is ok to say something about how at least with our strains and experimental protocols we find HNS silencing of csgD whereas Gerstel using a different whatever found HNS activation or whatever it is. It takes a level of brio to just claim ones results are more believable than others.*

We have softened this description and merely mention that our finding is different from theirs.

Reviewer #2:

*In this manuscript the authors build on previous work demonstrating that the SsrAB regulators are important to Salmonella typhimurium biofilm formation. They determine that the SsrA histidine kinase is not required, but that the SsrB response regulator is required for biofilm formation as measured using methylene blue staining, Congo Red, and Calcofluor plates. They then analyzed the colonies formed in a microfluidic chamber in M9 medium in a flow cell and in cholesterol coated eppendorf tubes to further substantiate the biofilm phenotype The authors provide support for the lack of phosphorylation of SsrB being what is necessary for SPI2 regulation of curli genes responsible for biofilms by using SsrB phosphorylation deficient bacteria, with mutations in SsrA and the ability to produce acetyl phosphate. The requirement of SsrB to form biofilms was repressed by deletion of H-NS providing in vivo evidence that SsrB could act by competing or silencing H-NS based repression. The authors then provide evidence that SsrB/HNS bind to the csg promoter responsible for the synthesis of curli. The results add to our knowledge of Salmonellae gene regulation of biofilm formation, but do not really define where and how this may occur in vivo and/or what specific environmental conditions would lead to activation but not phosphorylation of SsrB.*

Our previous work showed that SsrA and SsrB are uncoupled; i.e., at neutral pH, the SsrA kinase is present at very low levels and it is induced in the acidic macrophage vacuole. SsrB levels are constant at pH 7 and pH 5.6 (Feng et al., 2004). We discovered this over ten years ago and always wondered why the regulator was present without its kinase. This work answers that question by assigning a role for unphosphorylated SsrB. We have tried to make this point more clear in the revised manuscript.

*Specific comments:*

*A great deal of data was not shown and this should be corrected prior to publication.*

We respectfully disagree with the reviewer on this point. We have only one example of “data not shown” in the manuscript and it is that the SEM images of D56A SsrB are similar to wt. This was after showing many images that indicate D56A is identical to wt in biofilm behavior. We can put this data in supplemental information if the reviewers wish.

*Throughout the manuscript the authors claim that they showed different results than that shown in Hamilton et al.*

We mention this once, because of the questions that previous work raised, e.g. SsrA was not required but SPI-2 genes were down-regulated. This didn’t make sense, because SsrA/B up-regulates genes, this paradox led to the present investigation. We have re-worded our presentation of this work.

*However in this work the effect of an ssrA mutant on biofilm formation was complemented with a plasmid encoding both ssrA and ssrB. Therefore the results are likely a result of mutation of both genes or polarity.*

The previous study used a deletion of ssrA that was polar on the downsteam ssrB gene. This is now stated in the Discussion.

*Therefore the results are really similar and the authors have provided molecular detail as to the mechanism.*

*Since most of the paper is about mechanisms of transcription that lead to biofilm formation I am not sure what Figure 1 adds to the manuscript.*

In response to Reviewer #1, this figure has been removed.

*In Figure 2 the staining seems the same at 6 days. This is a surprising result, has the biofilm been formed that rapidly by another sensing mechanism?*

The staining is clearly not the same at 6 days in the *ssrB* null strain. We now show a different image that we hope makes this more obvious. It is clear that the wild type strain forms a network of typical mushroom-shaped macrocolonies by 6 d, by which time the *ssrB* strain has only managed to form lean aggregates.

*In Figure 2 why does the increase at later time points occur and why is it greater at 36 hours than at 47 and 69 hours? How many replicated experiments are represented by the figure and are the statistical analyses significant? This is not commented on in the figure legend.*

The *ssrB* strain is not completely defective in the formation of biofilms and similar to what we observe by confocal imaging of flow cell biofilms (Figure 1), at about three days the amount of biofilms do increase marginally in the mutant. This is expected as apart from SsrB, a range of environmental conditions via other global regulators also regulates the expression of *csgD*. However, the *ssrB* mutant forms less biofilms than the wild type across all time points. The statistical details were missed before and have been included now.

*The phenotype of the SsrA mutant should be shown in Figure 2.*

Since Reviewer #3 requested that we reduce the biofilm results to a single figure (see point 1, below), we elected not to add more data as requested here (we have shown by all the other methods that the *ssrA* mutant is like the *wt*).

*The higher migrating signal in the western blot in Figure 4 is said to be non-specific in the figure legend. However, It is not present to a very great degree in the wild type and is greater in the SseA and D56A mutant and absent in ssrB mutants. What is this band and does it indicate another larger form of CsgD or has a different promoter been utilized when non-phosphorylated SsrB is present? This band should be isolated and protein sequenced to define its nature.*

We obtained a monoclonal antibody and now the western blot is completely clean and it is clear that csgD is absent in the ssrB null strain (new Figure 3).

*What is the evidence that the binding angle observed in vitro with an artificial substrate by ATM reflects its ability to promote anti-silencing (subsection “Unphosphorylated SsrB binds to the csgD regulatory region”, first paragraph)?*

In our in vitro experiments, we found that SsrB bends DNA, resulting in highly curved DNA conformations. This DNA binding property is distinct from H-NS, which forms rigid nucleoprotein filaments and thus straight DNA conformations. Bent DNA is therefore an energetically unfavorable substrate for H-NS binding, and a likely mechanism of SsrB-mediated anti-silencing of H-NS repressed genes. In a mixture of 600 nM H-NS and 600 nM SsrB (Figure 6), we do not see evidence of sharply bent filaments. This is expected because H-NS dissociation is likely restricted to the filament ends. Such events occur due to the cooperative nature of H-NS binding that results in a chain of linked H-NS proteins. Hence, H-NS displacement by SsrB likely occurs progressively from the filament end. Indeed, we observed slow progressive H-NS filament dissociation in the presence of SsrB in single-molecule stretching experiments. We refer to Winardhi, Yan and Kenney (2015) for a more detailed discussion on H-NS anti-silencing by SsrB. Sentences to this effect have been added to the Discussion (subsection “The importance of anti-silencing in gene regulation”, last paragraph).

*Again condensation on a silica binding substrate of a regulatory region on an increase of concentration of the protein that binds may or may not be biologically relevant in vivo (subsection “Unphosphorylated SsrB binds to the csgD regulatory region”, last paragraph). The authors should reduce this speculation and consider performing the AFM in solution and under helium rather than nitrogen and see if similar results are obtained.*

We now include solution AFM, which shows the identical result as we presented previously (see Figure 6—figure supplement 1).

*The fact that HNS alters structure of DNA on a solid support does not reaffirm more than the biological data the mechanism by which HNS represses transcription, it just indicates under these conditions in binds to the DNA (subsection “SsrB binds an H-NS stiffened nucleoprotein filament at csgD”, first paragraph*).

As noted above, we see the same results with solution AFM (see Figure 6—figure supplement 1).

*I think the data support a model in which SsrB alters the bending of DNA, though it seems this may not be a direct effect in vivo and the authors should soften the statement that they have identified the mechanism by which unphosphorylated SsrB regulates csgD.*

We have >5 years’ worth of collaborative research with the Yan lab on H-NS and other nucleoid associated proteins that indicate the mechanism of gene silencing is via a stiffened nucleoprotein filament and relief of silencing occurs by DNA bending proteins such as SsrB (Walthers et al., 2011; Lim et al., 2012; Liu et al., 2010, Lim et al., 2012(a) and 2012(b) (recently reviewed in Winardhi, Yan and Kenney (2015)). We now cite more of that work in the Discussion (subsection “The importance of anti-silencing in gene regulation”).

*I am not sure the authors are correct in their ideas about SsrB function controlling extracellular lifestyle of Salmonellae since there are many growth conditions in which ssrAB are not expressed and the bacteria can form biofilms. Hence this may be most important for the expression of a biofilm under certain conditions of SPI2 expression but not SsrA activation as a kinase. This may actually occur inside host cells and recent evidence of cellulose expression in host cells and past work showing curli biosynthesis genes can be regulated by RpoS may suggest that they are expressed inside of host cells and hence the authors results may be most relevant to how curli may be expressed inside of host cells rather than SsrAB being another host switch more like PhoPQ.*

I am not sure what the reviewer is getting at here. Our previous work (Feng et al. 2004) and our current laboratory has examined SsrB and found it to be constitutive under all the conditions we tested (most recently using super-resolution imaging). So, we disagree with the reviewer on this point. We also do not know of any examples of SPI-2 expression without SsrA activation, since every SPI-2 gene that we have tested required SsrB~P for in vitro transcription (Walthers et al., 2007; Walthers et al., 2011).

*The title contains this “to P or not to P”, which I found confusing and relatively unscientific since it does not represent PO4. This should probably be changed prior to publication.*

Done.

*In Figure 6—figure supplement 1, why is so little shifting seen with SsrB in the absence of HNS? Is the affinity very low? Additional controls should include SsrB without the anti-ssrB antibody that causes the super shift. Is no binding seen without HNS? If the AFM is significant, binding should be seen without HNS present.*

We now include additional controls in the revised EMSA (see Figure 5—figure supplement 3 and Figure 6—figure supplement 3). In the AFM, SsrB binding and bending occurs at 120 nM, which is pretty high affinity for an unphosphorylated protein (e.g. OmpR binds to ompF and ompC with a Kd of 150 nM (Head et al., 1998). We also included a DNA binding mutant K179A (Carroll et al., 2009) and it clearly does not bind, so we demonstrate specificity as well. We also show that the EMSA complex formed by SsrB on *csgD_122_* can be dissociated by the addition of excess unlabeled *csgD_122_* fragment.

*The data would be strengthened by footprinting and defining the specific promotor sequences bound by SsrB and H-NS prior to publication.*

We now include the SsrB binding region based on the AFM (755 bp fragment) and the EMSA (122 fragment) taking into consideration the H-NS binding region (Gerstel et al., 2003), and all of our previous mapping and identifying of SsrB binding sites (Figure 6—figure supplement 4) (Walthers et al., 2007; Walthers et al., 2011).

Reviewer #3:

*The authors present a series of experiments demonstrating a novel role for a two component regulatory system, implicated in the virulence and persistence of Salmonella enteritica. The main novel aspects of the findings described in the manuscript are: 1. Demonstration that the SsrB response regulator is a molecular switch, reciprocally controlling two sets of traits associated with virulence (the expression of a type III secretion system) and biofilm formation; (2) Defining an active function for an un-phosphorylated response regulator; (3) Elucidation of a novel regulatory mechanism for a response regulator activity (anti-silencing) by DNA bending and (4) evolutionary implications for acquisition of a regulatory system by horizontal gene transfer, controlling the expression of core, chromosomal genes. Overall, this paper contains substantial amount of new results (in quality and in quantity) and represents information with broad interest to those readers who are interested in new molecular mechanisms of gene regulation, strategies of pathogenic organism capable of causing acute and chronic infections and their evolution. This is important work, established a new paradigm for two-component signaling. The only concern is with a few technical points related to certain experiments and a need for a few additional controls.*

We are thrilled to read that the reviewer understands the importance of our findings.

Reviewer #3 (Minor comments):

There are a few minor issue that the authors should address, mainly to make the manuscript more readable and highlighting its strongest points. 1) There is excessive amount of data on attempts to demonstrate that the absence of SsrB has an effect on biofilm formation. That could be simply reduced to a single figure comparing ssrB deletion with point mutants in SsrA and SsrB.

2) The major biofilm assay used in this work is based on adherence of bacteria to the sides of polystyrene tubes. This method actually measures only adhesion of the bacteria to their biofilm phenotype. A simple growth defect of the ssrB mutant would be also reflected as a decrease in adherence and erroneously concluded as a defect in biofilm formation. It is therefore necessary to somehow quantify the entire population of Salmonellae in the tube and assess what fraction of it adheres. A true biofilm-defective mutants should show a shift from a fraction that adhere to the non-adhering planktonic population.

Firstly, we present growth curves that indicate there is no growth defect (presented in Figure 1—figure supplement 1). Secondly, we assessed the adhering vs. non-adhering cells in each population (Mackenzie et al., 2015) (Figure 1—figure supplement 2 and Figure 1—figure supplement 3) and found that the adherent fraction was lower in the *ssrB* strain compared to the wild type, *ssrA* and D56A strains but, there were greater number of cells in the non-adherent sub-population for the *ssrB* mutant.

*3) Figure 4 is rather un-convincing; the lane containing the D56A mutant indicates some problems with transfer of CsgD from gels to the membrane and it does not correlate with the qRT-PCR data (Figure 4). The Western Blot should be either redone or left out.*

As mentioned above, we re-did the western blot with a monoclonal antibody and we now present it (Figure 3).

*4) The main issues with this work relates to the discovery of an anti-silencing mechanism by SsrB induced DNA bending. The elegant experiments using AFM need to be supported with certain controls, primarily aimed at demonstrating specificity of binding of SsrB to the csgD regulatory region (and lack of binding to other H-NS silenced regions) in both AFM and electrophoretic mobility shift assays.*

As mentioned above in response to Reviewer #2, these controls are now included for both AFM and EMSA. The K179A mutant clearly does not bind to DNA, demonstrating specificity (Figure 5 and Figure 5—figure supplement 3).

References cited:

1) Russo, F. D., Slauch, J. M. & Silhavy, T. J. Mutations that affect separate functions of OmpR the phosphorylated regulator of porin transcription in Escherichia coli. J. Mol. Biol. 231, 261-273 (1993).

2) Lim, C. J., Whang, Y. R., Kenney, L. J. & Yan, J. H-NS paralogue StpA represses by forming a rigid protein scaffold along DNA that blocks DNA accessibility Nuc. Acids. Res. 40:3316-3328, 3316-3328 (2012).

3) Lim, C. J., Whang, Y. R., Kenney, L. J. & Yan, J. Gene silencing H-NS paralogue StpA forms a rigid protein filament along DNA that blocks DNA accessibility. Nucleic acids research 40, 3316-3328, doi:10.1093/nar/gkr1247 (2012).

4) Head, C. G., Tardy, A. & Kenney, L. J. Relative binding affinities of OmpR and OmpR-phosphate at the ompF and ompC regulatory sites. J. Mol. Biol. 281, 857-870 (1998).